# Harmonizing Stochasticity and Determinism: Scene-responsive Diverse Human Motion Prediction

**Zhenyu Lou**[1]    **Qiongjie Cui**[2*]

Tuo Wang[4]    Zhenbo Song[2]    Luoming Zhang[1]    Cheng Cheng[5]    Haofan Wang[3]
Xu Tang[3]    Huaxia Li[3]    Hong Zhou[1]

[1]Zhejiang University, [2]Nanjing University of Science and Technology, [3]Xiaohongshu Inc,
[4]University of Texas at Austin, [5]Concordia University,
`11915044@zju.edu.cn`   `cuiqiongjie@126.com`

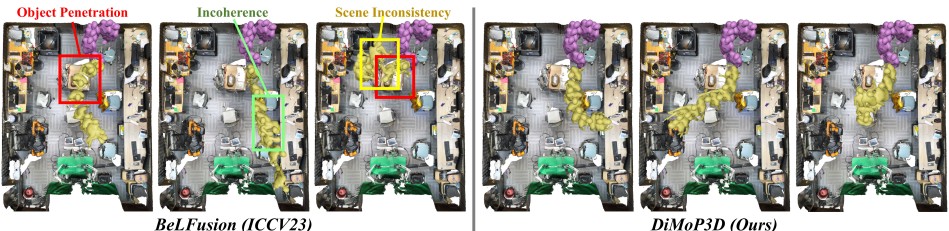

Figure 1: Comparison of our DiMoP3D with the SoTA baseline [5]. Purple meshes represent observations, and yellow meshes denote predictions. `DiMoP3D` produces high-fidelity, diverse sequences tailored to real-world 3D scenes, while BeLFusion's inadequate scene context integration leads to issues such as object penetration, motion incoherence, and scene inconsistency.

## Abstract

Diverse human motion prediction (HMP) is a fundamental application in computer vision that has recently attracted considerable interest. Prior methods primarily focus on the stochastic nature of human motion, while neglecting the specific impact of the external environment, leading to the pronounced artifacts in prediction when applied to real-world scenarios. To fill this gap, this work introduces a novel task: predicting diverse human motion within real-world 3D scenes. In contrast to prior works, it requires harmonizing the deterministic constraints imposed by the surrounding 3D scenes with the stochastic aspect of human motion. For this purpose, we propose `DiMoP3D`, a diverse motion prediction framework with 3D scene awareness, which leverages the 3D point cloud and observed sequence to generate diverse and high-fidelity predictions. `DiMoP3D` can comprehend the 3D scene and determine the probable target objects and their desired interactive pose based on the historical motion. Then, it plans the obstacle-free trajectories toward these interested objects and generates diverse and physically consistent future motions. On top of that, `DiMoP3D` identifies deterministic factors in the scene and integrates them into stochastic modeling, making the diverse HMP in realistic scenes become a controllable stochastic generation process. On two real-captured benchmarks, `DiMoP3D` has demonstrated significant improvements over state-of-the-art methods, showcasing its effectiveness in generating diverse and physically consistent motion predictions within real-world 3D environments. More details and the video demo are available at the webpage `https://sites.google.com/view/dimop3d`.

---

*Corresponding author

# 1 Introduction

Human motion prediction (HMP), *i.e.,* forecasting future human poses based on observation, is crucial for applications including autonomous vehicles and human-robot collaboration [15, 20, 39, 44, 54, 60, 66]. Many existing works [1, 22, 40, 49, 76, 89] formulate HMP as a deterministic problem, aiming to generate a single future sequence. However, it fails to capture the inherent stochasticity of human motion, where multiple plausible outcomes can arise from a single observation. Recent research has shifted towards diverse or stochastic HMP, to achieve multiple plausible predictions [3, 6, 30, 59, 72], which holds the potential in real-world applications and is the focus of our work.

Recent advances in diverse HMP primarily focus on stochastic predictions, where a random factor from the latent space conditions the diversity of predictions alongside observed motions [3, 12, 30, 59, 71, 72, 82]. While these methods predict multiple plausible futures from a single past motion, they typically disregard the 3D environment, operating within an idealized, context-free framework. This limitation becomes apparent in real-world applications, where motion must conform to physical and semantic scene constraints [29, 67, 68, 78, 90], leading to issues like obstacle penetration and unrealistic interactions, as in Figure 1. This gap underscores the need for a new task that merges diverse HMP within real-world 3D scenes, enhancing both practicality and applicability.

Recognizing existing limitations, this work introduces a novel task, making diverse HMP within real-world 3D scenes. Its objective is to break the previous idealized context-free setup towards a realistic and practical setting, which involves several key challenges: **(1) Harmonizing Stochasticity and Determinism:** This task necessitates a delicate harmonization between the stochastic nature of human motion and the deterministic constraints from 3D scenes, thereby broadening the scope of traditional HMP; **(2) Scene-Motion Intermodal Coordination:** It requires analyzing coordination between human motion and scene dynamics to align predictions with contextual elements, which involves identifying human intentions and potential interactive objects; **(3) Behaviorally Coherent Physical Consistency:** The predictions must adhere to deterministic constraints, including physical consistency (*e.g.,* avoid collision) and behavior coherence (*e.g.,* sitting on a chair, not lying).

To tackle these challenges, we introduce `DiMoP3D` (**Di**verse **M**otion **P**rediction in **3**D Scenes), a framework for generating diverse, physically consistent, and plausible human motion predictions in real-world 3D scenes, which comprises three main components: (1) Context-aware Intermodal Interpreter analyzes potential areas of human interest and goals, essentially intentions within a scene. Our method enhances traditional scene understanding by integrating 3D point clouds with observed motions for context-aware intermodal analysis. It first encodes the point cloud, segments object instances, and then pinpoints potential interaction targets, emphasizing intended factors while filtering out less probable ones. This strategy further transforms the task of diverse HMP into a controllable stochastic generation process; (2) Behaviorally-consistent Stochastic Planner then constructs behaviorally consistent action plans, representing stochastic conditional factors. Recognizing that human interaction with specific objects often follows deterministic patterns, we prioritize predicting the final human pose upon reaching each target, and generate obstacle-free trajectories toward it; (3) Self-prompted Motion Generator harmonizes the stochastic nature of human motion with deterministic constraints in a self-prompted manner to produce varied predictions based on the conditional factor. To ensure coherence with this factor, it employs a denoising diffusion model, guiding the motion denoising process toward a deterministic, obstacle-free final state.

Our contributions are threefold: (1) We introduce a novel and challenging task of predicting diverse human motions within real-world 3D scenes, advancing beyond the traditional scope of diverse HMP from an idealized context-free setting to a more realistic and practical one. (2) We propose `DiMoP3D` to tackle this task. It harmonizes the deterministic constraints of 3D scenes with the stochastic nature of human motion, enabling diverse and plausible motion predictions in real-world scenarios. (3) Evaluations on two real-captured benchmarks, GIMO and CIRCLE, show that `DiMoP3D` significantly outperforms existing SoTA methods, particularly in terms of physical consistency.

# 2 Related Work

## 2.1 Stochastic Human Motion Prediction

Human motion prediction diverges into deterministic and stochastic methods. Deterministic models aim to predict a singular sequence that closely aligns with future movements [18, 19, 40, 49, 76],

yet often encounter quality degradation over longer time spans ($>$ 1-$sec$) due to the stochastic nature of human movement. In contrast, stochastic HMP [2, 3, 5, 44, 82] embraces this variability by generating a range of plausible future motions and modeling the distribution of human behaviors. Notably, it also encompasses the most probable future sequence targeted by deterministic models, as a likely scenario within its broader distribution. Such methodologies enhance applications across autonomous driving [7, 33, 43], patient care [10, 42], and human flow prediction [32, 36], by embracing a wider range of potential scenarios and have become a focal point of contemporary research.

Dominant approaches include VAEs, GANs, flow networks, and diffusion models [3, 6, 12, 30, 38, 59, 71, 72, 82]. Despite promising progress in modeling stochastic human motions, a critical challenge persists: human motion is not only stochastic but also heavily influenced by the external environment. In real-world settings, human motion is intricately intertwined with surrounding scenes, necessitating that future trajectories comply with the scene's physical constraints (*e.g.,* avoid object penetration). Additionally, predicted motions should be semantically consistent with the expected human-object interactions (*e.g.,* sit for a chair, lie for a bed). Addressing these requirements calls for a sophisticated approach to diverse HMP that balances the stochastic aspect of human motion with the deterministic factors imposed by the surrounding 3D scenes, which is the focus of our work.

## 2.2 3D Scene Encoding and Scene-aware Motion Prediction

3D scene understanding is essential in various applications, prompting extensive research on representations like RGB-D maps [9, 52, 61], scene graphs [77, 79, 84], 3D voxels [23, 48, 74], and notably, 3D point clouds [55, 56, 65, 87]. Recognizing the importance of 3D scene information in human motion prediction [14, 21, 26, 35, 80], our work integrates 3D point clouds as the scene representation due to their direct derivation from sensing technologies.

To enhance fidelity of HMP in real-world scenarios, the connection between human actions and scene context has made scene-aware motion generation a major research focus [8, 17, 28, 88]. Early methods [8, 17, 67] relied on object bounding boxes, 2D images, and depth maps, which are insufficient for capturing real 3D environments. Recent advances use 3D point clouds for scene representation [29, 69, 88]: GIMO [88] employs a bidirectional transformer to fuse human motion and scene features, [50] predicts future contact maps, and [46] extracts global and local salient points.

However, these methods predict a single sequence [4, 50, 88], whereas our approach models a distribution of potential outcomes. Some stochastic methods add diversity, but [8] relies on 2D inputs, limiting 3D interactions, and [28] uses predefined objects without inferring targets. In contrast, our model parses the 3D scene through cross-modal, object-specific human interest analysis and predicts scene-aware motions in real 3D scenarios, greatly enhancing adaptability and authenticity.

## 2.3 Motion Synthesis in 3D Scenes

Recent advancements in paired scene-motion data [4, 27, 88] have sparked a new research direction in synthesizing motions in 3D scenes [29, 41, 47, 53, 69, 78]. Specifically, SAMP [28] employs a conditional variational autoencoder (cVAE) to generate one frame per forward pass within a three-stage stochastic pipeline, COUCH [86] introduces a human-chair dataset with an autoregressive, contact-satisfying method, and DN-Synt [68] proposes a hierarchical framework for effective scene-aware human motion synthesis. With the rise of language models, methods utilizing natural language prompts have emerged [16, 69, 75, 85]. HUMANISE [69] proposes an attention-based language-prompted synthesis method. Additionally, diffusion models have shown significant promise [16, 29, 37, 63, 64, 70]. Notably, AffordMotion [70] employs scene affordance as an intermediate representation, achieving state-of-the-art performance in object interaction synthesis.

Our novel task of scene-aware diverse HMP is partly inspired by advances in motion synthesis but is tailored for distinctly different applications and challenges. This task is further distinguished by key innovations: (1) Temporal-Dependent Prompting. Contrary to motion synthesis, which often lacks temporal context and follows static user instructions, our approach conditions predictions on the unique prompt of historical human motion, enabling autonomous inference of human intentions. (2) Context-Aware Scene Parsing. Moving beyond traditional scene understanding, our method infers potential movement targets through the integration of context-aware intermodal insights. Utilizing scene determinism to enhance the fidelity of diverse HMP predictions.

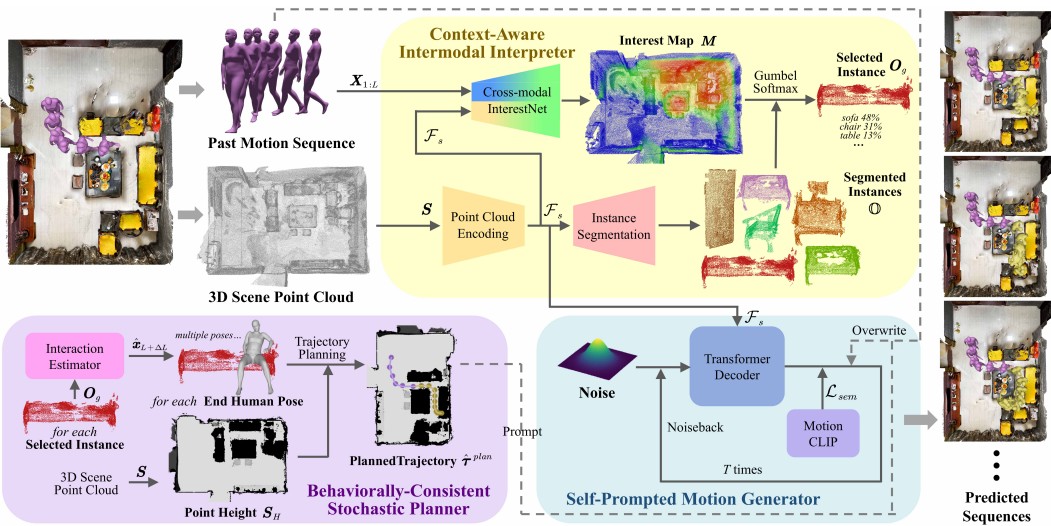

Figure 2: The architecture of DiMoP3D. DiMoP3D incorporates two modalities of input, the past motion and the 3D scene point cloud. Initially, the Context-aware Intermodal Interpreter encodes the point cloud to features $\mathcal{F}_s$, identifies interactive objects $\mathbb{O}$, and uses a cross-modal InterestNet to pinpoint potential interest areas, sampling a target instance $\boldsymbol{O}_g$ according to interest map $\boldsymbol{M}$. Following this, the Behaviorally-consistent Stochastic Planner forecasts the interactive human end-pose $\hat{\boldsymbol{x}}_{L+\Delta L}$, and devises an obstacle-free trajectory $\hat{\boldsymbol{\tau}}^{plan}$ towards this pose. The sampled end-pose and trajectory are incorporated as a stochastic conditional factor to prompt the Self-prompted Motion Generator to generate physically consistent future motions.

## 3 Method

### 3.1 Problem Setup

Given $L$ historical human poses $\boldsymbol{X}_{1:L} = \{\boldsymbol{x}_1, \boldsymbol{x}_2, \cdots, \boldsymbol{x}_L\}$ within 3D scenes represented by point clouds $\boldsymbol{S} \in \mathbb{R}^{n_p \times 6}$, our goal is to predict $K$ different scene-consistent future motions $\{\hat{\boldsymbol{X}}_{L:L+\Delta L}^{(i)}\}_{i=1}^K$. Here, $n_p$ denotes the number of points, each described by 3D spatial coordinates and RGB color information. $L$ and $\Delta L$ denote the lengths of the observed and predicted sequences, respectively. Each pose is described as the SMPL-X representation $\boldsymbol{x}_l = (\boldsymbol{t}_l, \boldsymbol{o}_l, \boldsymbol{p}_l)$ [88], where $\boldsymbol{t}_l \in \mathbb{R}^3$ denotes the global translation, $\boldsymbol{o}_l \in SO(3)$ denotes the orientation, and $\boldsymbol{p}_l \in \mathbb{R}^{32}$ refers to the body pose embedding. We set $L = 3\text{-}sec$ and $\Delta L = 5\text{-}sec$ to achieve a long-term prediction [40, 49]. Then, the task can be formulated as:

$$P(\hat{\boldsymbol{X}}_{L:L+\Delta L}|\boldsymbol{X}_{1:L}, \boldsymbol{S}) = \int \max_{\theta} P(\hat{\boldsymbol{X}}_{L:L+\Delta L}|\boldsymbol{X}_{1:L}, \boldsymbol{S}, \theta) P(\theta|\boldsymbol{X}_{1:L}, \boldsymbol{S}) d\theta. \tag{1}$$

The stochastic process $\theta$ is sampled jointly from the past motion $\boldsymbol{X}_{1:L}$ and the scene $\boldsymbol{S}$, and then utilized to condition the prediction motion $\hat{\boldsymbol{X}}_{L:L+\Delta L}$. We propose DiMoP3D to solve this novel task, which involves the following novelties: **Context awareness**: unlike traditional diverse HMP methods [12, 59, 71] that focus solely on human motion, our task is more challenging as it requires harmonizing the stochastic nature of human motion with the deterministic constraints of the surrounding 3D scenes. **Autonomous intention estimation**: different from motion synthesis, our task requires independent intention estimation based on past motion, to prompt future motion prediction.

### 3.2 Context-Aware Intermodal Interpreter

Scene information plays a crucial role in predicting future motion [8, 17, 67]. Despite the stochastic manner of human motion, it is still feasible to deduce human interests and likely goals within a scene. For instance, in Figure 2, the door behind the person is unlikely to be the target based on their trajectory away from it, while the sofa, coffee table, and distant chair may emerge as potential points of interest. Diverging from traditional scene parsing methods, our approach integrates a scene-motion intermodal coordination to better suppose human intentions in real-world settings.

In machine vision, 3D point clouds sourced directly from sensing devices have become fundamental for scene representation [55, 65, 87] and serve as the input for our scene interpreter. Acknowledging the significant influence of past motions and scene context on future human movements, we emphasize the need for a context-aware intermodal analysis that integrates historical motion with scene point clouds to infer potential human intentions. Furthermore, since human movements typically involve interactions with target objects [11, 24, 81], our approach identifies objects within the scene and computes a human interest score for each, rather than analyzing isolated points. This helps determine specific interactive targets, making the motion prediction process more controllable and enabling diverse prediction by sampling different interactive targets.

To achieve this, our scene interpreter employs a UNet-like encoder-decoder architecture [58], comprising an instance segmenter for object recognition and an interest net for human interest inference. These two modules share the same point cloud encoder for efficiency but use different decoders, as illustrated in the yellow box in Figure 2. The shared encoder processes and downsamples the point cloud $\boldsymbol{S}$ into a compact feature representation $\mathcal{F}_s \in \mathbb{R}^{n_p' \times c}$, with $n_p' \ll n_p$ ($8 \leq n_p' \leq 50$ in our cases), and $c$ represents the feature dimension. Subsequently, two decoders are employed to segment objects $\mathbb{O}$, and predict human interests $\boldsymbol{M}$ in the scene, respectively:

$$\mathcal{F}_s = \text{SceneEncoder}(\boldsymbol{S}), \quad \mathbb{O} = \text{InstanceSegmenter}(\mathcal{F}_s), \quad \boldsymbol{M} = \text{InterestNet}(\mathcal{F}_s, \boldsymbol{X}_{1:L}). \quad (2)$$

Here, $\mathbb{O} = \{\boldsymbol{O}_1, \boldsymbol{O}_2, ..., \boldsymbol{O}_{n_o}\}$ denotes the set of $n_o$ segmented objects, each being a subset of the scene pointcloud ($\boldsymbol{O}_i \subseteq \boldsymbol{S}$). $\boldsymbol{M} \in \mathbb{R}^{n_p \times 1}$ denotes the per-point interest map, with higher values indicating a greater likelihood of targeting specific scene elements. Once $\boldsymbol{M}$ is obtained, we compute the probability $P_i$ of each object $\boldsymbol{O}_i \in \mathbb{O}$ being selected as an interaction target:

$$M_i = \sum \boldsymbol{M}[p] / len(\boldsymbol{O}_i), \quad p \in \boldsymbol{O}_i, \quad (3)$$

$$\{P_1, P_2, ..., P_{n_o}\} = \text{Softmax}(\{M_1, M_2, ..., M_{n_o}\}; \phi), \quad (4)$$

where $len(\boldsymbol{O}_i)$ denotes the number of points in $\boldsymbol{O}_i$, $\boldsymbol{M}[p]$ is the interest value of point $p$, and $\phi = 0.5$ represents the temperature factor that controls randomness in sampling. During inference, we sample the target object $\boldsymbol{O}_g (g \in \{1, 2, ..., n_o\})$ based on the probability distribution $\{P_1, P_2, ..., P_{n_o}\}$.

In cases where there is no human-object interaction or objects are beyond reach within $\Delta L$, no explicit target object exists. To handle this, we include the ground as a potential target, voxelized into smaller patches to improve granularity. Each ground patch, treated as an individual object, has a side length of $s = 0.5$ meters to balance accuracy and efficiency. The interest scores and interaction probabilities for ground patches are then calculated similarly to other objects, as in Eq.3 and Eq.4.

Our scene interpreter aligns observed motions with scene context, filtering out less likely engagement areas and identifying potential targets. We note that this approach makes the diverse HMP controlled by those deterministic elements of the scene, thereby enhancing the physical consistency of predictions. Representing human intention through scene-motion intermodal analysis, the selected target object $\boldsymbol{O}_g$ directs the subsequent planning process, as outlined below.

### 3.3 Behaviorally-Consistent Stochastic Planner

Ensuring collision-free and scene-consistent human behavior is crucial but often overlooked, while learning these patterns directly requires impractically large datasets. To tackle these challenges, we employ an action planner to plan obstacle-free trajectories toward the target $\boldsymbol{O}_g$ and deterministically predict the interactive human end-pose $\hat{\boldsymbol{x}}_{end}$ associated with $\boldsymbol{O}_g$. These intermediate predictions serve as stochastic conditional factors, guiding to craft future motions that respect physical constraints and typical human-environment interactions while incorporating motion diversity.

To enable effective navigation and collision avoidance, we utilize a scene height map $\boldsymbol{S}_H \in \mathbb{R}^{n_s \times 1}$ to delineate accessible areas and detect obstacles, inspired by [68, 73]. We first compute obstacle-free trajectories toward the target object $\boldsymbol{O}_g$ using a modified A* algorithm (details on generating diverse trajectories are in Appendix B), and then employ a single-layer transformer $\psi$ to predict per-frame human velocity, sampling discretize points from the planned trajectory:

$$\boldsymbol{\tau}^{plan} = \text{A*}(\boldsymbol{S}_H; \boldsymbol{X}_{1:L}, \hat{\boldsymbol{x}}_{end}), \quad (\hat{\boldsymbol{\tau}}^{plan}, t_{end}) = Sample(\boldsymbol{\tau}^{plan}, \psi(\boldsymbol{X}_{1:L})). \quad (5)$$

Here, $\boldsymbol{\tau}^{plan}$ denotes the continuous trajectory, $\hat{\boldsymbol{\tau}}^{plan} = \{\hat{\boldsymbol{t}}_{L+1}^{plan}, \hat{\boldsymbol{t}}_{L+2}^{plan}, ..., \hat{\boldsymbol{t}}_{L+\Delta L}^{plan}\}$ represents the sampled discretize trajectory points, and $t_{end}$ is the estimated timestamp for the end of the interactive

motion. Since the length of this trajectory varies across sequences, the human may not reach the target object exactly at the prediction horizon $\Delta L = 5$-$sec$. In these cases, if the target object is too distant to reach within $\Delta L$ ($t_{end} > \Delta L$), we truncate the trajectory $\boldsymbol{\tau}^{plan}$ to fit within $\Delta L$ and adjust the target to the nearest ground patch. Conversely, if the target is reached too early ($t_{end} < \Delta L$), we keep the subject relatively static after $t_{end}$. This adjustment ensures that the planned trajectory aligns with the prediction horizon. We then discretize the continuous trajectory to obtain the per-frame global translation $\hat{\boldsymbol{\tau}}^{plan}$ to guide subsequent motion generation:

$$\hat{\boldsymbol{\tau}}^{plan} = \{\hat{\boldsymbol{t}}_{L+1}^{plan}, \hat{\boldsymbol{t}}_{L+2}^{plan}, ..., \hat{\boldsymbol{t}}_{L+\Delta L}^{plan}\}. \tag{6}$$

In addition to physical constraints, human interactions with specific objects often follow deterministic patterns despite potential action diversity. For instance, "set," and "wipe," are reasonable actions for a table, whereas "sit" and "lie" are not. Traditional diverse HMP methods typically overlook these Human-Object Interaction (HOI) patterns, leading to motion and scene inconsistencies. Differing from these methods, our approach predicts the target object $\boldsymbol{O}_g$ in advance, enabling us to predict the interactive HOI end-pose $\hat{\boldsymbol{x}}_{end}$ before the full-sequence prediction:

$$\hat{\boldsymbol{x}}_{end} = \text{HOI-Estimator}(\boldsymbol{O}_g). \tag{7}$$

This end-pose represents the final state of the prediction, secures appropriate human interaction with the scene. To this end, our planner constructs an obstacle-free trajectory, and the predicted end-pose (along with $t_{end}$) as a stochastic conditional factor $\theta$ from a scene-motion intermodal perspective:

$$\theta = (\hat{\boldsymbol{\tau}}^{plan}, \hat{\boldsymbol{x}}_{end}, t_{end}). \tag{8}$$

This factor further prompts the motion generator to predict behaviorally consistent motions.

## 3.4 Self-Prompted Motion Generator

By constructing a stochastic conditional factor $\theta$ in advance, DiMoP3D operates as a self-prompted motion generator, which harmonizes the stochastic factor with deterministic motion generation rooted in $\theta$. To generate diverse predictions that closely align with the predicted conditional factor $\theta$, we utilize a motion diffuser, taking advantage of the diffusion model's ability to effectively guide intermediate results [12, 59, 71]. Additionally, to further maintain semantic coherence and physical consistency, we propose a semantic alignment inspector to supervise the denoising process.

For simplicity, we denote the sequence at noising step $t$ as $\boldsymbol{X}^t$. Diffusion is modeled as a Markov noising process $\{\boldsymbol{X}^t\}_{t=0}^T$, with $\boldsymbol{X}^0$ drawn from the data distribution, and:

$$q(\boldsymbol{X}^t|\boldsymbol{X}^{t-1}) = \mathcal{N}(\sqrt{\alpha_t}\boldsymbol{X}^{t-1}, (1-\alpha_t)\boldsymbol{I}). \tag{9}$$

Here $\alpha_t \in (0, 1)$. When $\alpha_T$ approaches 0, we approximate $\boldsymbol{X}_{1:L+\Delta L}^T \sim \mathcal{N}(\boldsymbol{0}, \boldsymbol{I})$, where $\boldsymbol{0}$ and $\boldsymbol{I}$ represent the zero matrix and the identity matrix, respectively. To effectively integrate scene and motion features in our predictions, our diffusion model $\mathcal{M}_D$ employs a transformer decoder to model distribution akin to the reversed diffusion process, leveraging its capacity for cross-modal attention. Instead of predicting noise, we predict the clean sample directly, following [63, 64, 83]:

$$\bar{\boldsymbol{X}}^0 = \mathcal{M}_D(\hat{\boldsymbol{X}}^t, \mathcal{F}_s, t), \tag{10}$$

where $\bar{\boldsymbol{X}}^0$ represents the intermediate denoising result at each denoising step.

To align the predicted sequence with the observed $\boldsymbol{X}_{1:L}$, the planned trajectory $\hat{\boldsymbol{\tau}}^{plan}$, and the forecasted end-pose $\hat{\boldsymbol{x}}_{end}$ at time $t_{end}$, we adjust the corresponding segments after each denoising step. To be sepecific, for each frame $\bar{\boldsymbol{x}}_i^0 = (\bar{\boldsymbol{t}}_i^0, \bar{\boldsymbol{o}}_i^0, \bar{\boldsymbol{p}}_i^0)$:

$$\bar{\boldsymbol{x}}_i^0 = \begin{cases} \boldsymbol{x}_i^0 & \text{if } i \le L, \\ (\hat{\boldsymbol{t}}_i^{plan}, \bar{\boldsymbol{o}}_i^0, \bar{\boldsymbol{p}}_i^0) & \text{if } L < i < L + t_{end}, \\ \hat{\boldsymbol{x}}_{L+\Delta L} & \text{if } i + t_{end}, \\ \bar{\boldsymbol{x}}_i^0 & \text{if } i > L + t_{end}, \end{cases} \quad \text{for each frame } i. \tag{11}$$

This modified prediction is then noised back before the next denoising step:

$$\hat{\boldsymbol{X}}^{t-1} = \mathcal{N}(\sqrt{\alpha_t}\boldsymbol{X}^{t-1}, \gamma_t \boldsymbol{I}), \tag{12}$$

where $\gamma_t$ represents the posterior variance based on $\alpha$ at step $t$. Upon completing $T$ denoising steps, the motion generator yields a cohesive sequence $\hat{\boldsymbol{X}}^0$, which integrates smoothly with the observed sequence and aligns with the planned trajectory and goals.

To enhance the consistency between the predicted motion and the target object, we introduce a semantic alignment inspector leveraging MotionCLIP [62]. It computes a HOI semantic loss via natural language descriptions as follows:

$$\mathcal{L}_{sem} = \frac{1}{L + \Delta L} \sum_{l=1}^{L+\Delta L} \left( 1 - \cos\Big(\text{MC}_{Motion}(\hat{\boldsymbol{X}}_{l:L+\Delta L}^{t}), \text{MC}_{Text}(\mathbb{D}_g)\Big) \right). \tag{13}$$

Here, $\text{MC}_{Motion}, \text{MC}_{Text}$ denote MotionCLIP's motion and text encoders, respectively, with $\mathbb{D}_g$ signifying the interaction description template related to the class of the sampled target object $\boldsymbol{O}_g$. Acknowledging that HOI predominantly occurs later in motion sequences, our semantic loss formulation weights later motion frames more heavily to accurately capture these interactions.

# 4 Experiment

## 4.1 Experimental Setup

**Dataset-1: GIMO** [88], which records motion sequences represented by full-body SMPL-X poses with $\approx 129K$ frames. It consists of 14 scenes with 3D point clouds, each scene is captured by a 3D LiDAR sensor, containing 10-20 objects with $\approx 500K$ vertices. For a fair comparison, we follow the official split to divide the dataset into training and testing sets according to the scenes.

**Dataset-2: CIRCLE** [4] comprises 10 hours of high-fidelity full-body motion sequences from 5 subjects across nine apartment scenes. Utilizing a Vicon system with 12 cameras at 120 FPS and the AI Habitat VR environment for virtual world simulation, CIRCLE achieves precise motion and scene capture. It offers an integrated apartment mesh, designating each room as an individual scene. The dataset encompasses motion sequences for 128 tasks, totaling over 7,000 sequences.

We also notice other related datasets [27, 28, 50], yet find limitations precluding their use (*e.g.*, jittering, sequence length, absence of human meshes).

**Baselines.** Our DiMoP3D is compared with four contemporary methods: DLow [82], SmoothDMP [72], BeLFusion [5], and BiFU [88]. DLow [82] applies a flow network, SmoothDMP [72] is VAE-based, and BeLFusion [5] is diffusion-based, which achieves SoTA performance in diverse HMP. These three, however, do not focus on scene-aware diverse HMP, setting BiFU [88], a deterministic scene-aware method, apart as an essential control for our analysis.

**Metrics.** To align with existing literature that evaluates human skeleton metrics, we utilize the SMPL model [45] to convert body parameters $\boldsymbol{X}_{1:L+\Delta L}$ into skeletons $\boldsymbol{J}_{1:L+\Delta L} = \{\boldsymbol{j}_1, \boldsymbol{j}_2, ..., \boldsymbol{j}_{L+\Delta L}\}$, where each $\boldsymbol{j}_i \in \mathbb{R}^{n_j \times 3}$ represents a skeleton with $n_j = 22$ joints, following [25].

DiMoP3D is evaluated for diversity, accuracy, and physical consistency in scene-aware predictions. We begin with the well-established pipeline in [82]: Prediction diversity is quantified using the Average Pairwise Distance **(APD)** by computing the L2 distance across predicted sequences. The Average Displacement Error **(ADE)** measures the reconstruction accuracy among the whole predicted sequence, while the Final Displacement Error **(FDE)** measures accuracy of the furthest frame, alongside their multimodal counterparts, **MMADE** and **MMFDE**, for diverse HMP scenarios.

To measure the physical consistency of the predicted motion within 3D scenes, an additional metric, the Average Cumulated Penetration Depth **(ACPD)** is introduced, following [78, 83]:

$$\text{ACPD}(\boldsymbol{J}) = \frac{1}{\Delta L} \sum_{l=L+1}^{L+\Delta L} \sum_{n=1}^{n_j} max\Big( -\text{SDF}(\boldsymbol{j}_l[n], \boldsymbol{S}), \ 0 \Big), \tag{14}$$

where $\boldsymbol{j}[n]$ denotes the position of the $n$-th joint in the skeleton, and $\text{SDF}(\cdot, \boldsymbol{S})$ refers to the signed distance function [51] of the scene point cloud $\boldsymbol{S}$. For training details, please refer to Appendix A.

Table 1: Comparison of DiMoP3D with baselines on GIMO [88] and CIRCLE [4] datasets. The best outcomes are highlighted in bold. Given that BiFU [88] employs a deterministic prediction approach, diversity metrics such as APD, MMADE, and MMFDE are not applicable.

| | Method | APD ↑ | ADE ↓ | FDE ↓ | MMADE ↓ | MMFDE ↓ | ACPD ↓ |
|---|---|---|---|---|---|---|---|
| GIMO [88] | Dlow [82] | 55.12 | 13.70 | 16.88 | 15.96 | 17.31 | 14.55 |
| | SmoothDMP [72] | **68.80** | 11.17 | 14.51 | 13.67 | 15.42 | 15.08 |
| | BeLFusion [5] | 38.04 | 9.69 | 11.19 | 11.28 | 12.02 | 13.73 |
| | BiFU [88] | – | 7.11 | 8.39 | – | – | 3.73 |
| | **DiMoP3D** | 48.30 | **5.66** | **6.81** | **6.57** | **7.44** | **0.98** |
| CIRCLE [4] | Dlow [82] | 49.37 | 11.70 | 14.46 | 13.49 | 14.95 | 12.52 |
| | SmoothDMP [72] | **57.38** | 9.75 | 12.04 | 11.31 | 13.57 | 13.92 |
| | BeLFusion [5] | 39.46 | 7.91 | 9.30 | 9.56 | 10.04 | 14.06 |
| | BiFU [88] | – | 5.80 | 6.99 | – | – | 2.11 |
| | **DiMoP3D** | 42.24 | **5.09** | **6.12** | **5.95** | **6.48** | **0.87** |

Table 2: Ablation of four main components in DiMoP3D over the sequences of the GIMO [88].

| Ablation | APD ↑ | ADE ↓ | FDE ↓ | MMADE ↓ | MMFDE ↓ | ACPD ↓ |
|---|---|---|---|---|---|---|
| w/o InterestNet | 52.63 | 6.17 | 7.48 | 7.20 | 8.09 | 1.00 |
| w/o HOI-Estimator | 46.97 | 5.95 | 7.27 | 6.72 | 7.84 | 1.53 |
| w/o TrajectoryPlanner | **57.29** | 6.39 | **6.81** | 7.28 | 7.45 | 3.29 |
| w/o SemanticInspector | 47.79 | 5.82 | 6.82 | 6.75 | 7.46 | 1.06 |
| **DiMoP3D** | 48.30 | **5.66** | **6.81** | **6.57** | **7.44** | **0.98** |

## 4.2 Main Results

Table 1 demonstrates DiMoP3D's superiority over the baseline methods across nearly all evaluation metrics on both datasets. The non-scene-aware methods (Dlow, SmoothDMP, BeLFusion) exhibit limited motion accuracy (ADE, FDE, MMADE, MMFDE) and physical scene consistency (ACPD), which we hypothesize is due to (1) lack of scene awareness, resulting in notable inconsistency in real-world applications, and (2) the absence of explicit motion goals, which hinders precise long-term (5-*sec*) motion forecasting. Despite their higher scores in diversity (APD), this is attributed to their erratic and unpredictable predictions, disregarding the scene context (detailed in Sec 4.4).

DiMoP3D's enhanced performance stems from three key factors: (1) Diversity. The stochastic conditional factor introduces diversity through multiple mechanisms: the intermodal interpreter sets broad motion objectives, the stochastic planner generates a variety of end-poses and trajectories, and the motion generator achieves diverse motion poses. This multi-faceted approach ensures a breadth of plausible actions are considered, enabling DiMoP3D to achieve a considerable APD score. (2) Accuracy. DiMoP3D outperforms every baseline in ADE, FDE, MMADE, and MMFDE for a large margin, even the deterministic BiFU. By estimating future human action based on a scene-motion intermodal analysis, DiMoP3D implicitly infers the subject's intent. This boosts the probability of accurately identifying the subject's genuine intent as the basis for prediction, thereby improving the prediction precision. The combination of accurate intermodal scene interpreting and stochastic planning ensures precise motion prediction for each sequence. (3) Physical consistency. Our motion generator employs a diffusion model, prompted by the predicted stochastic factors. It also ensures motion coherence through priors overwrite at each denoising step. This dual focus on deterministic constraints enables DiMoP3D to achieve superior motion-scene consistency.

This superior performance demonstrates the efficacy of our DiMoP3D in predicting diverse human motion in 3D scenes, as further evidenced by subsequent ablation studies and visualizations.

## 4.3 Ablation Studies

In Table 2, we dissect the impact of excluding four pivotal components from DiMoP3D. First, eliminating InterestNet markedly decreases performance across ADE, FDE, MMADE, and MMFDE (first row). This decline stems from the process of selecting the target object $O_g$ from $\mathbb{O}$, which reverts to random sampling without scene-motion crossmodal analysis, impairing DiMoP3D's ability to deduce human intentions. Consequently, the accuracy of predicting real actions diminishes, highlighting the significance of integrating multimodal scene-motion analysis for scene-aware HMP.

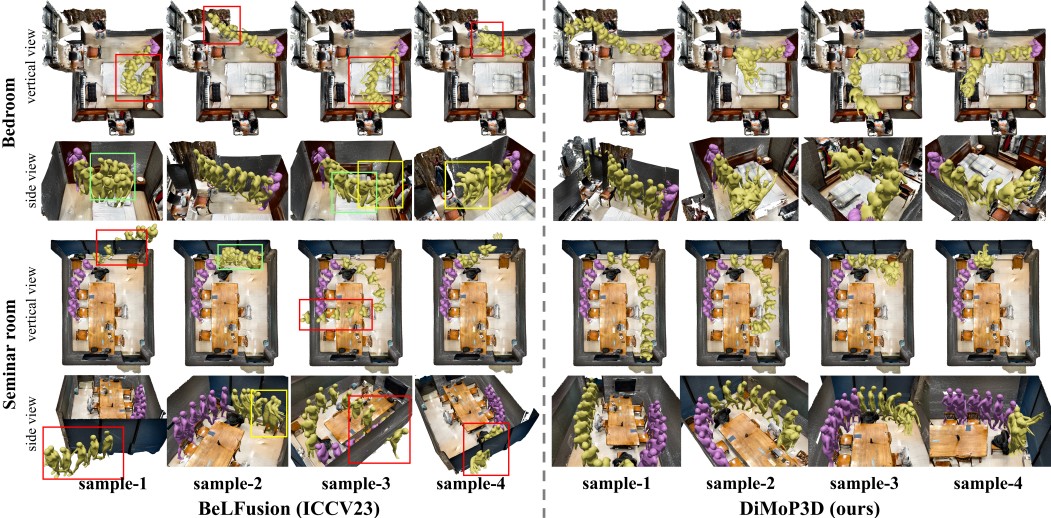

Figure 3: Visual comparisons between DiMoP3D and SoTA BeLFusion in bedroom and seminar room scenarios. BeLFusion's predictions, which rely solely on past human motion without considering 3D scene context, are shown on the left. In contrast, DiMoP3D, displayed on the right, incorporates interactive goals and designs obstacle-free trajectories for each sequence. Purple meshes depict observed motions, while yellow ones signify predicted future motions. For clarity, distortions in BeLFusion's predictions are marked: red boxes for object penetration, green boxes for motion incoherence, and yellow boxes for scene inconsistency.

Addressing the role of stochastic planner, its absence undermines the planning of actions, including the prediction of end-poses by the HOI-Estimator and trajectory planning via the A* TrajectoryPlanner. Without these two components, the motion generator struggles to predict end-poses or future trajectories with scene consistency. Notably, omitting the HOI-Estimator results in imprecise interactive end-poses, often causing the subject to intersect with the target object in later frames, as evidenced by increased ACPD and reduced FDE and MMADE (second row). Similarly, excluding the TrajectoryPlanner significantly elevates ACPD (third row), indicating frequent subject penetrations into the scene context while approaching the end-pose. These findings underscore the vital role of coordinated end-pose and trajectory prediction in predicting motion within 3D scenes effectively.

Finally, the SemanticInspector enhances the scene-motion alignment through natural language, with its omission resulting in higher ADE and MMADE. Please refer to Appendix C for further ablations.

### 4.4 Visualization

To delve deeper into DiMoP3D, in Figure 3, we showcase DiMoP3D's predictions across two scenarios, contrasting them with the SoTA baseline BeLFusion [5].

In bedroom scenario, the subject stands still behind the door. BeLFusion's predictions show notable issues with object and wall penetrations. Furthermore, sample-1 and 3 are marked by abrupt, illogical movements, and sample-3 and 4 display glaring scene inconsistencies: sample-3 has the subject sitting on the bare floor, and sample-4 involves the subject reaching for non-existent items. Conversely, DiMoP3D ensures physical consistency, directing each prediction towards a specific movement goal: opening a window, lying on the bed, accessing a cabinet, and sitting on a chair.

In the seminar room scene, the subject is moving forward. BeLFusion struggles again, with sample-1, 3, and 4 depicting the subject unrealistically exiting the room, and sample-2 showing an inconsistent motion of picking up a curtain. DiMoP3D, however, delivers high-fidelity predictions, depicting the subject walking through a door, grasping items, sitting, and pulling down curtains, respectively.

To emphasize DiMoP3D's predictive diversity, we further visualize various end-poses generated by the HOI-Estimator in Figure 4. Overall, DiMoP3D consistently delivers diverse, realistic, and physically-consistent motion predictions with clear objectives, benefiting from our conditional factor prediction schema which models human-object interactions and navigates obstacle-free trajectories.

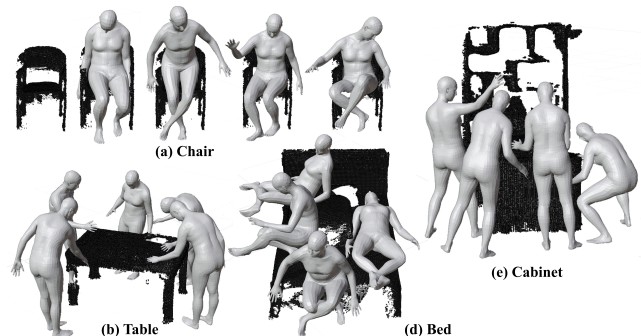

Figure 4: Visualizations of diverse predicted end-poses across five object point clouds. The HOI-Estimator can generate a variety of human-object interactive poses tailored to specific scenarios.

Table 3: Comparison of DiMoP3D with scene-aware motion synthesis methods.

| Method | APD ↑ | ADE ↓ | FDE ↓ | MMADE ↓ | MMFDE ↓ | FID ↓ | ACPD ↓ |
|---|---|---|---|---|---|---|---|
| SAMP [28] | 31.73 | 9.83 | 9.28 | 11.16 | 10.13 | 1.493 | 1.69 |
| DN-Synt [68] | 44.60 | 9.71 | 7.16 | 11.29 | 7.83 | 1.026 | 1.21 |
| AffordMotion [70] | **52.54** | 8.96 | 8.14 | 10.38 | 8.95 | **0.687** | 1.26 |
| **DiMoP3D** | 48.30 | **5.66** | **6.81** | **6.57** | **7.44** | 0.769 | **0.98** |

## 4.5 Compared with Motion Synthesis Methods

In this section, we compare our DiMoP3D with three scene-aware motion synthesis approaches on GIMO [88]: SAMP [28], DN-Synt [68], and AffordMotion [70]. SAMP and DN-Synt utilize VAE architectures, while AffordMotion employs a diffusion-based model.

To adapt these methods for our diverse HMP task, we initialize motion synthesis from the last observed frame $x_L$ and encode the complete observed sequence $X_{1:L}$ into a unified embedding for historical motion conditions using a 2-layer transformer encoder, similar to the embedding technique in [57]. Additionally, we introduce the FID metric, commonly used in motion synthesis, to evaluate the discrepancy between the distributions of generated and original dataset motions.

The results in Table 3 reveal an intriguing pattern: synthesis methods exhibit higher ADE than FDE. This occurs because, although these methods include explicit end-pose estimators yielding accurate final pose predictions, they struggle to condition on past motion. Consequently, they produce motion incoherence and significant prediction errors along the trajectory from the observation to the final pose. Notably, AffordMotion achieves the best APD and FID scores, which we attribute to its design that prioritizes fidelity over accuracy and allows a higher degree of freedom. Meanwhile, DiMoP3D also demonstrates competitive performance in these metrics. These findings underscore DiMoP3D's capability to harmonize the stochastic nature of human motion and the deterministic constraints from the scene and the past motion, achieving superior performance in diverse scene-aware HMP, and maintaining competitive diversity and fidelity even when compared to SoTA synthesis method.

## 5 Conclusion and Limitation

This work introduces a novel task of predicting diverse human motion in 3D scenes, along with a novel framework, DiMoP3D, to address it. By incorporating multimodal motion-scene analysis, DiMoP3D identifies areas or objects the subject is likely to interact with, enabling diverse, accurate, and physically consistent human motion prediction. Evaluated on the GIMO and CIRCLE datasets, DiMoP3D reduces ADE and FDE by nearly half compared to the state-of-the-art baseline BeLFusion, while maintaining high physical consistency. These results underscore the importance of scene awareness in diverse human motion prediction for real-world applications.

Despite its strong performance, DiMoP3D predicts motion in a fixed sequence length. When the actual sequence length differs, it either keeps the subject relatively static or truncates the sequence. Future work could explore variable-length motion prediction or end-to-end prediction, where the motion generator predicts sequence length and generates motion simultaneously.

## Acknowledgements

This work was supported in part by the National Key Research and Development Program of China (2022YFC3602601), in part by the Natural Science Foundation of Jiangsu Province (BK20220939), and in part by the National Natural Science Foundation of China (62306141).

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

# A  Traning Details

Training DiMoP3D encompasses separate phases for InterestNet, HOI-Estimator, and the self-prompted motion generator. This section delineates the specific training methodologies for each component. All training is conducted on a single NVIDIA RTX3090 GPU, with the complete pipeline converging in $\sim 8$ hours.

**Training InterestNet.** We adopt the ScanNet [21] pretrained SoftGroup model as our encoder and segmenter to enhance performance, and a transformer decoder for the interest net to achieve cross-modal analysis. The original datasets [4, 88] lack annotations for the interest map $M$; hence, we enrich them with such annotations for each motion clip. InterestNet is employed to elucidate the relationship between the observed motion and the scene, facilitating the prediction of human interests, including likely destinations and objects of interaction. To ensure the interest map accurately represents these aspects, we annotate the human interest map $M$ based on three critical factors: the contact area between humans and objects ($M_{cont}$), the proximity of humans to scene elements ($M_{dist}$), and the spatial relationship between each object $O_i$ and the target object $O_g$ ($M_{obj}$):

$$M_{cont} = f_{\mathcal{N}(0,\sigma_1)}(\text{Dist3D}(S, p_{cont})), \tag{15}$$

$$M_{dist} = \frac{1}{\Delta L} \cdot \sum_{l=L+1}^{L+\Delta L} \frac{(l-L)^2}{(\Delta L)^2} \cdot f_{\mathcal{N}(0,\sigma_2)}(\text{Dist2D}(S, t_l)), \tag{16}$$

$$M_{obj} = \sum_{O_i \in \mathbb{O}} f_{\mathcal{N}(0,\sigma_3)}(\text{Dist2D}(O_i, O_g)) \cdot O_i, \tag{17}$$

$$M = \lambda_{cont} M_{cont} + \lambda_{dist} M_{dist} + \lambda_{obj} M_{obj}. \tag{18}$$

Here, $p_{cont}$ denotes the human-object contact position, $f_{\mathcal{N}(\mu,\sigma)}(\cdot)$ denotes the Gaussian function with mean $\mu$ and standard deviation $\sigma$. Smaller $\sigma$ concentrates the interest map while larger $\sigma$ distributes attention more broadly. We set $\sigma_1 = 0.3$ and $\sigma_2 = \sigma_3 = 1.0$ for balance. Dist3D$(\cdot)$ and Dist2D$(\cdot)$ are the 3D and X-Z 2D distance functions (Y-axis denotes height). Hyperparameters $\lambda_{cont} = 3, \lambda_{dist} = 10, \lambda_{obj} = 1$ are adjusted to maintain a balance among the factors.

Upon annotating the interest map $M$, InterestNet is trained using a KL divergence loss to minimize the discrepancy between the distributions of the predicted interest map and the annotated $M$:

$$\mathcal{L}_{\text{Interest}} = KL(M \,||\, \text{InterestNet}(\mathcal{F}_s, X_{1:L})). \tag{19}$$

**Training HOI-Estimator.** The HOI-Estimator employs an autoregressive conditional variational autoencoder (cVAE) [34] architecture, designed to predict a range of feasible end-poses. It predicts interactive end-poses $\hat{x}_{end}$ based on the target object's point cloud $O_g$. Initially, we centralize the object on the X-Z plane (assume the Y-axis denotes height) by normalizing its position to the origin:

$$O_g^{Norm} = O_g - \frac{1}{len(O_g)} \sum_{p \in O_g} p_{xz}, \tag{20}$$

where $len(O_g)$ indicates the point count in $O_g$ and $p_{xz}$ is the X-Z coordinates of point $p$. The HOI-Estimator utilizes a conditional Variational Autoencoder (cVAE) [34] for encoding-decoding:

$$\mu, \sigma = \text{Encoder}(O_g^{Norm}, X_{1:L}), \tag{21}$$

$$\hat{x}_{end} = \text{Decoder}(O_g^{Norm}; \mu, \sigma). \tag{22}$$

The HOI-Estimator is trained to minimize the L2 loss between the reconstructed interactive pose $\hat{x}_{end}$ and the ground-truth $x_{end}$:

$$\mathcal{L}_{\text{HOI}} = ||\hat{x}_{end} - x_{end}||_2^2. \tag{23}$$

**Training motion generator.** The motion generator is pre-trained on the HumanML3D dataset [25]. We predict the clean sample $\bar{X}^0$ directly instead of predicting noise following [63, 64, 83], consisting of the basic diffusion loss:

$$\mathcal{L}_{\text{base}} = ||\bar{X}^0 - X||_2^2, \tag{24}$$

Table 4: Description template of semantic inspector among 18 objects in the ScanNet dataset [21].

| [OBJ] | [ACT] | | | |
|---|---|---|---|---|
| ground | walks on | stands on | | |
| cabinet | opens | searches | reaches hands to | picks things from |
| bed | sits on | lies on | | |
| chair | sits on | | | |
| sofa | sits on | | | |
| table | sets | wipes | reaches hands to | picks things from |
| door | opens | closes | reaches hands to | passes |
| window | opens | closes | reaches hands to | |
| bookshelf | arranges | leans on | reaches hands to | picks things from |
| picture | hangs | takes down | reaches hands to | |
| counter | wipes | leans on | reaches hands to | |
| desk | wipes | organizes | reaches hands to | picks things from |
| curtain | opens | closes | reaches hands to | |
| refrigerator | opens | closes | reaches hands to | picks things from |
| shower curtain | opens | closes | reaches hands to | |
| toilet | sits on | flushes | reaches hands to | |
| sink | washes | | reaches hands to | picks things from |
| bathtub | flushes | | reaches hands to | takes a shower in |
| other furniture | uses | checks | reaches hands to | interacts with |

supplemented by the semantic alignment loss (detailed in Section 3.4 of the main paper), forming the total training loss for motion generator:

$$\mathcal{L}_{\text{diff}} = \mathcal{L}_{\text{base}} + \mathcal{L}_{\text{sem}}. \tag{25}$$

**Description Template of Semantic Inspector.** The semantic inspector incorporates language description to supervise the predicted motion to be consistent with target objects. To enable this supervision, we design a language description template for each class of objects in the format of:

$$\text{The person walks forward then } [ACT] \text{ the } [OBJ], \tag{26}$$

where $[ACT]$ and $[OBJ]$ are placeholders for the action and object, respectively. There are a total of 18 classes of objects (excluding wall and floor) in the pre-trained ScanNet dataset [21], and we design corresponding $[ACT]$ for each class, detailed in Table 4.

## B  Modified A* Trajectory Planner

To facilitate diverse trajectory prediction, we enhance the conventional A* trajectory planner, allowing for the iterative generation of valid paths while penalizing previously traversed positions.

Initially, we generate the scene height map $\boldsymbol{S}_H$ as described in Alg. 1. To streamline height analysis and A* pathfinding, we voxelized the scene point cloud $\boldsymbol{S}$ along the X and Z axes (with the Y-axis representing height) into a grid of 0.02-meter resolution. We then identify the maximum height within each grid cell, omitting ceilings and tall cabinets, which, despite their height, do not impede movement and would otherwise be inaccurately marked as obstacles.

Leveraging the generated height map $\boldsymbol{S}_H$, we outline our modified dynamic A* trajectory planner in Alg. 2. Our method transcends traditional one-hot encoding for marking obstacles by enabling navigation over lower barriers through a continuous cost function derived from $\boldsymbol{S}_H$. We adopt a power function to model the cost, with cell height acting as the exponent. To ensure smoother trajectories and reduce sudden changes, an L2 penalty on angular velocity is integrated into the cost framework. Additionally, to prevent the selection of repetitive paths, we increase the cost of cells once traversed. The modified A* algorithm then iteratively generates paths until the cumulative cost exceeds a predefined threshold or the maximum path count is attained, as in Figure 5.

## C  Additional Ablation Studies

**Scene feature for baselines.** To assess the influence of scene features on baseline methods, we conduct supplemental experiments integrating scene features into these methods. Despite the baseline models mentioned in the main paper [5, 72, 82] not inherently accommodating scene features, we

---

**Algorithm 1** get_heightmap(S):

---

```
# S: Scene point cloud with shape (Np, 3), where Np is the point count

invalid = -10000, grid = 0.02, h_th = 1.2
Sg = VoxelGridXZ(S, grid)        # X,Z coordinates are gridded, while Y is not
H = ones((Sg.maxx-Sg.minx)/grid, (Sg.maxz - Sg.minz)/grid) * invalid
For p in Sg:
    If p.y < h_th:        # Excluding the ceiling and high cabinets
        x, z = Sg.grid(p.x, p.z)
        H[x, z] = max(Hmap[x, z], p.y)
return H
```

---

---

**Algorithm 2** dynamic_astar(X, d, H):

---

```
# X: Observed motion trajectory with shape (L, 2), where L is the input length
# d: Destination position with shape (2)
# H: Scene height map with shape (lenX, lenZ)

def cost_function(lastcost, p, path):
    return lastcost + C[p] + angular_vel(path)

invalid = -10000, grid = 0.02, h_th = 1.2, noise = 1.0
costbase = 1000, costth = 1000, costinc = 10
H[H==invalid] = h_th
C = costbase ** H
paths = []
For npath in range(MAX_PATHS):
    path, cost = astar(C + randlike(C)*noise, X[-1], d, cost_function)
    If cost > mincost + costth: break
    C[path] += costinc
    paths.append(bessel_smoothing(path))
return paths
```

---

Table 5: Results of appending scene features for the baseline methods on GIMO [88].

| Method | scene | APD↑ | ADE↓ | FDE↓ | MMADE↓ | MMFDE↓ | ACPD↓ |
|---|---|---|---|---|---|---|---|
| Dlow [82] | | 55.12 | 13.70 | 16.88 | 15.96 | 17.31 | 14.55 |
| SmoothDMP [72] | | **68.80** | 11.17 | 14.51 | 13.67 | 15.42 | 15.08 |
| BeLFusion [5] | | 38.04 | 9.69 | 11.19 | 11.28 | 12.02 | 13.73 |
| Dlow [82] | ✓ | – | – | – | – | – | – |
| SmoothDMP [72] | ✓ | 66.36 | 11.02 | 14.49 | 13.43 | 15.37 | 14.71 |
| BeLFusion [5] | ✓ | 36.89 | 9.55 | 11.04 | 11.12 | 11.76 | 13.52 |
| **DiMoP3D** | ✓ | 48.30 | **5.66** | **6.81** | **6.57** | **7.44** | **0.98** |

augmented their input by appending the scene feature $\mathcal{F}_s$ along the temporal dimension for [5, 82], and along the joint dimension for [72], to explore potential performance.

The results in Table 5 reveal that Dlow failed to convergence, likely attributed to its recursive architecture, wherein the concatenated scene features diverge significantly from the distribution of motion features, making the network difficult to learn. Both SmoothDMP and BeLFusion exhibit marginal improvements, suggesting that the direct integration of scene features into the motion generator yields limited effectiveness. Conversely, these results underscore the efficacy of DiMoP3D's strategy of predicting the conditional factor, highlighting its superiority in harmonizing the scene and the observed motion.

**Ablation on the scene segmenter.** To evaluate DiMoP3D's robustness across various point cloud instance segmentation methods, we performed additional experiments with different segmentators. Higher-quality segmenters yield more precise object delineations, enabling the selection of more accurate targets and reducing object boundary violations. Table 6 illustrates that DiMoP3D main-

Table 6: Results of DiMoP3D with various scene segmenters. "mAP50" denotes the mean average precision at 50 IoU threshold for each segmenter on the ScanNetv2 dataset [21]. Higher "mAP50" represents better segmenter performance.

| Segmentator | mAP50 | APD↑ | ADE↓ | FDE↓ | MMADE↓ | MMFDE↓ | ACPD↓ |
|---|---|---|---|---|---|---|---|
| PointGroup [31] | 63.6 | 46.22 | 5.70 | 6.93 | 6.63 | 7.56 | 1.05 |
| HAIS [13] | 69.9 | 46.97 | 5.67 | 6.87 | 6.61 | 7.50 | 1.02 |
| SoftGroup [65] | 76.1 | **48.30** | **5.66** | **6.81** | **6.57** | **7.44** | **0.98** |

Table 7: Results of DiMoP3D with various trajectory planners. Trajectory planner has no effect on FDE and MMFDE as they are end-pose errors, so we omit them in this table.

| Planner | APD↑ | ADE↓ | MMADE↓ | ACPD↓ |
|---|---|---|---|---|
| Naive A* | 39.26 | 5.93 | 6.82 | 0.98 |
| ours w/o continuous cost | 45.59 | 5.77 | 6.72 | 0.98 |
| ours w/o angular penalty | **50.17** | 5.91 | 6.88 | **0.97** |
| **ours** | 48.30 | **5.66** | **6.57** | 0.98 |

tains stable performance even when the quality of the segmentation method declines, underscoring its robustness to variations in segmentation.

**Ablation on the trajectory planner.** To validate the significance of our enhanced A* trajectory planner, we conducted an ablation study. Results presented in Table 7 show that the naive A* method underperforms due to its deterministic nature. Excluding the continuous cost model results in a binary scene representation, disregarding traversable lower obstacles and thus reducing path diversity. Additionally, omitting the angular velocity penalty leads to paths with abrupt turns, detracting from prediction accuracy. These outcomes significant the critical role of our modified A* trajectory planner in achieving nuanced and reliable path predictions.

# D More Visualizations

**Instance segmentation and interest map.** Exploring our multimodal scene parser, we present visualizations of 3D scene instance segmentation alongside the corresponding interest maps $M$ for three distinct samples. Figure 6 (upper) showcases the segmented instances within the scene, while Figure 6 (lower) evidences our multimodal InterestNet's capacity to deduce potential human intentions.

In the bedroom setting, with the person initially stationary and then beginning to move forward, the future action remains ambiguous. Consequently, almost all objects are highlighted as potential targets in $M$, except for the door situated behind the individual. In the living room scene, as the person navigates the narrow gap between a chair and a table, the sofa, an additional chair, and the table emerge as points of interest. Conversely, the proximate chair and a distant door garner lesser attention, aligning with the observed motion pattern. A similar observation is noted in a laboratory environment.

These results underscore our multimodal scene parser's adeptness at inferring potential human intentions, and the generated interest map $M$ can accurately reflect anticipated human interests and interactions.

**Multiple samples with single target.** To further explore DiMoP3D's capacity for predicting diverse motions toward a deterministic target object, additional visualizations are presented.

The left panel of Figure 7, showcases predicted sequences where the person navigates different paths to reach the destination, performing varied actions such as looking down and reaching toward the target. The right panel illustrates sequences of the person adopting different positions for lying and sitting on the bed. These results affirm DiMoP3D's adeptness at generating varied and coherent human motions aimed at a specific target object, further ensuring diversity.

**Comparison with synthesis methods.** To further explore the performance of synthesis methods in diverse scene-aware HMP tasks, we present visual comparisons between DiMoP3D and the SoTA synthesis method AffordMotion in Figure 8. AffordMotion's predicted sequences tend to remain relatively static at the onset of prediction, which we attribute to its method of smoothly synthesiz-

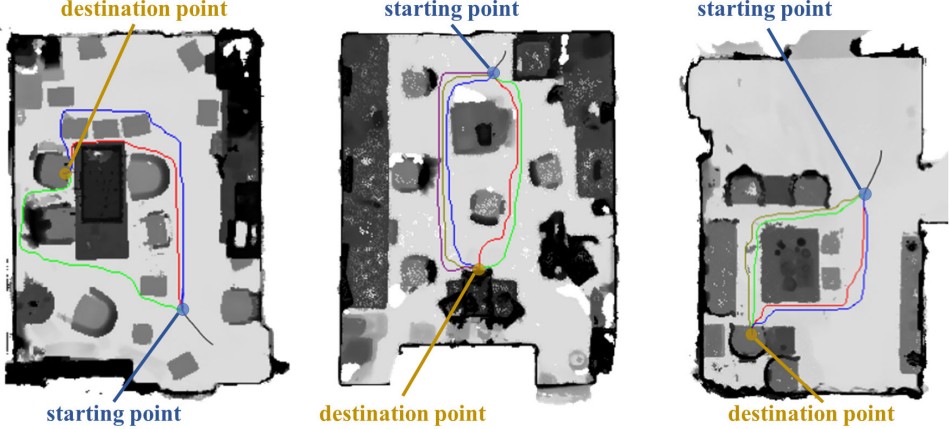

Figure 5: Visualization samples of the modified A* trajectory planner. Black lines denote the observed trajectory, while colored lines represent the generated paths.

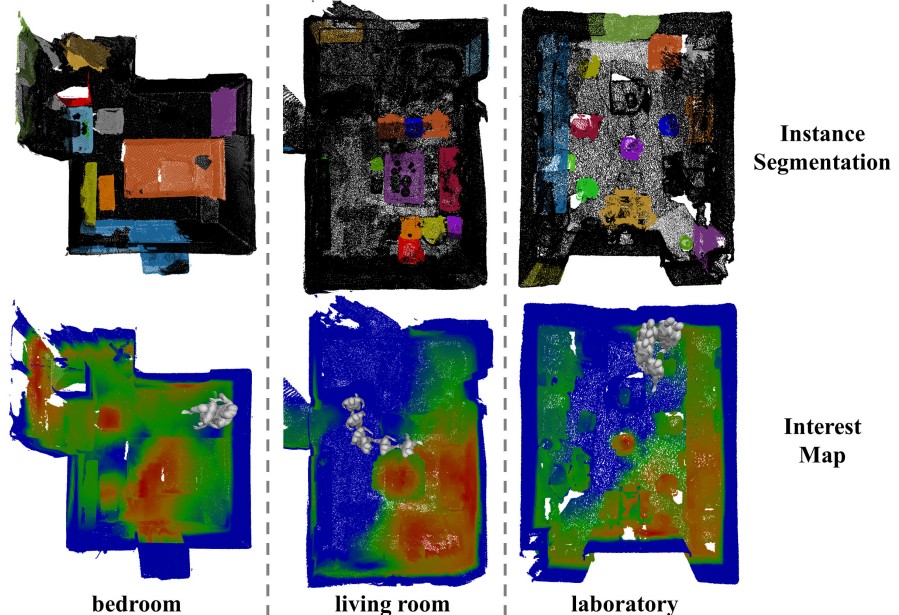

Figure 6: Visualization of 3D scene instance segmentation (upper) and the corresponding interest map (lower). Red points in the interest map denote higher interest, while blue points denote lower interest. Leveraging the insight provided by the predicted interest map enables the exclusion of improbable or illogical targets, thereby enhancing the reliability and scene congruency of predictions.

ing human motion irrespective of prior motion states, resulting in significant errors along the paths. Particularly in samples 3 and 4, AffordMotion shows notable motion incoherence, characterized by abrupt changes at the transition between prediction and observation. In contrast, DiMoP3D considers past motion, predicting coherent and plausible sequences, thereby demonstrating its superior performance in diverse scene-aware HMP tasks. While motion synthesis methods show promise in handling diverse scene-aware HMP, they face the critical challenge of integrating deterministic cues from past motions effectively.

## E    Potential Broader Impacts

The proposed DiMoP3D introduces a novel diverse scene-aware motion prediction framework, which may involve the following broader impacts:

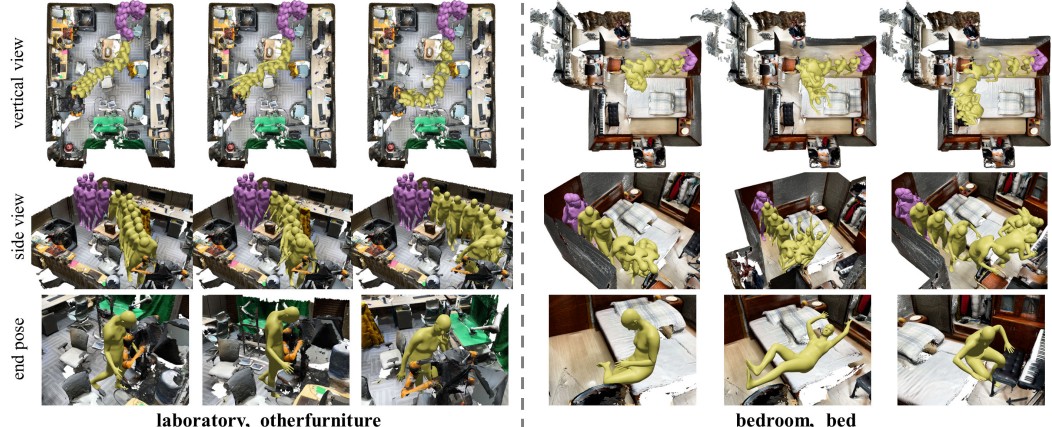

Figure 7: Visualization of multiple samples with fixed target object. DiMoP3D is able to predict motions with diverse trajectories and actions (or end poses) toward a deterministic target object, while maintaining each motion sequence to be consistent with the observation and the scene.

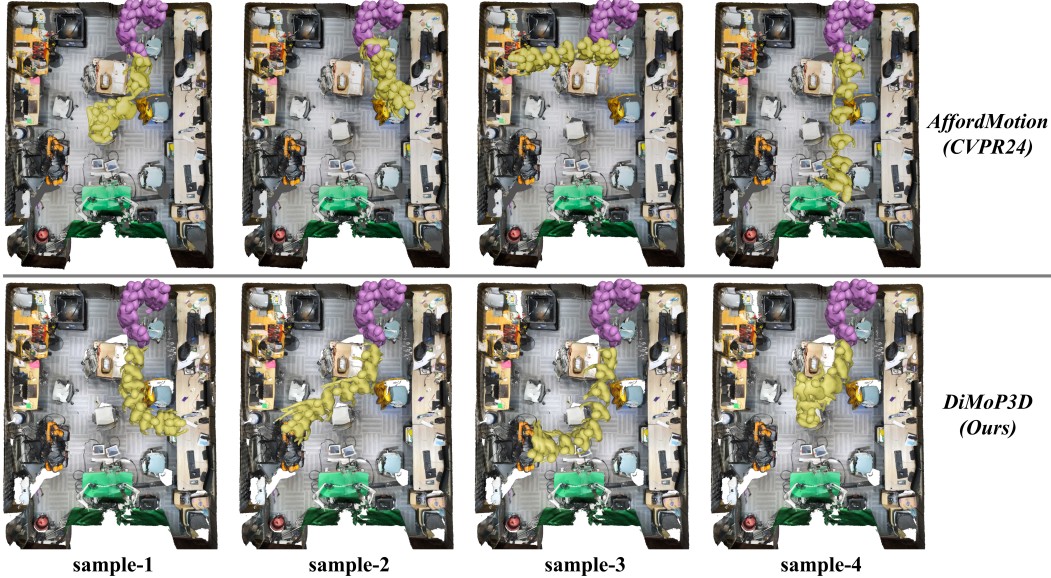

Figure 8: Visualization comparisons between DiMoP3D and SoTA synthesis method AffordMotion. The results from AffordMotion demonstrate significant motion incoherence.

- **Enhanced Safety in Robotics and Automation.** DiMoP3D can improve the interaction between humans and robots. By predicting human motions accurately, robots can avoid collisions and unsafe interactions, making environments safer for both humans and machines.

- **Improved VR and Gaming Experiences.** In VR and video games, this framework can lead to more realistic and responsive interactions with virtual characters and environments. By understanding and predicting how a human might move within a scene, VR systems can offer more immersive and natural experiences, enhancing user engagement and satisfaction.

- **Advancements in Assistive Technologies.** For people with disabilities or the elderly, assistive technologies equipped with human motion prediction can anticipate needs or actions (like falling or reaching for an object) and provide timely assistance or interventions, thereby enhancing independence and quality of life.

- **Applications in Autonomous Vehicles.** Integrating this framework into autonomous vehicle systems can improve pedestrian safety and traffic management. By predicting human movements, autonomous vehicles can better navigate complex urban environments where interactions with pedestrians are frequent and unpredictable.

