# OpenReview forum: "Harmonizing Stochasticity and Determinism: Scene-responsive Diverse Human Motion Prediction"
_NeurIPS.cc/2024/Conference — NeurIPS 2024 poster_

### Official Review · Reviewer_Q59f · 2024-06-25

**Soundness:** 4
**Presentation:** 4
**Contribution:** 4
**Rating:** 8
**Confidence:** 3

**Summary:**

The paper present a method for human motion prediction that take into account the environment (3d point cloud) in which the person moves. The method work with several steps: first from the past motion an area of interest is estimated and several object of interest (bed, chair...) with which the person might interact. after selecting the most likely object the method will estimate an interaction with the object (e.g sitting on the bed or lying on the bed) and plan a trajectory to reach the object. Finally with the trajectory and the estimated interaction the model generate the 3d motion from the end of the past motion to the end of the interaction. Since this is a novel task the author compare to two type of method : motion prediction and environment aware motion synthesis and outperform the state of the art on both tasks.

**Strengths:**

The paper introduce a new task of environment aware motion prediction where the environment is a static 3d point cloud.

Interesting approach where the models first find the most likely item to be used then create a trajectory and finally generate the future motion based on that trajectory.

The paper provides extensive and detailed experiments and ablation study on two datasets

The qualitative results look very good.

Despite tackling a new task the authors compare with sota methods for two related tasks.

The paper is clear,  well written and well detailed.

**Weaknesses:**

At least some results of the scene aware motion synthesis from the appendix should be moved to the main paper. Especially since there is no qualitative comparison with other scene aware method in the main paper.

In figure 1 it is not clear that diffusion is being used.

A qualitative comparison with bifu would have been interesting since its results are the best after the proposed method.

**Questions:**

Videos of the generated motion would have been appreciated

line 261 introduces => introduced

**Limitations:**

Limitations and impacts are discussed in the appendix.

---

> ### Author Rebuttal · Authors · 2024-08-07
>
> **Q1**: Synthesis methods move to the main paper.
>
> **A1**: We will incorporate the discussion of the synthesis method into the main body of the paper in the next revision. Specifically, Section B.1 will be moved to Section 2.3, B.2 will be positioned in a new subsection between Sections 4.2 and 4.3, and B.3 will be relocated to Section 4.4. Additionally, we will expand our discussion on the differences between tasks in the introduction and in Section 3.1, and we will elaborate on the experimental results of the synthesis method in the newly added subsection of Section 4.
>
> **Q2**: In figure 1 it is not clear that diffusion is being used.
>
> **A2**: Figure 1 showcases the diverse predictions of DiMoP3D. Diffusion model adapts better to our diverse prediction task and outperforms both cVAE and GAN in terms of performance. Its stochastic nature also enhances the stochastic factors in DiMoP3D. If you are referring to Figure 2, we detail the structure of DiMoP3D there, illustrating how diffusion is implemented within the Self-Prompted Motion Generator. We will enhance the clarity of the use of diffusion model in Figure 2 in the next version.
>
> **Q3**: Qualitative results with BiFU.
>
> **A3**: We have included a qualitative comparison with BiFU in Section-5 of the anonymous page (see Appendix. A, line-588). Notably, DiMoP3D not only generates diverse predictions but also achieves closer alignment with the ground truth than BiFU in sequences with the same targets. This accuracy is facilitated by the guidance of predicted interactive poses and trajectories.
>
> **Q4**: Videos.
>
> **A4**: Kindly refer to the anonymous page in Appendix. A (line-588).
>
> **Q5**: Typos in line 261.
>
> **A5**: We will correct the typo in the next version.
>
> [1] Zheng, et al. "Gimo: Gaze-informed human motion prediction in context." ECCV2022.

---

> > ### Comment · Reviewer_Q59f · 2024-08-09
> > **rating after rebuttal**
> >
> > the rebuttal answered most of my and the other reviewers concerns, I keep my original rating.

---

> > > ### Author Response · Authors · 2024-08-11
> > > **Comment from authors**
> > >
> > > Happy to solve your concerns, and thank you for the time and effort!

---

### Official Review · Reviewer_wnWw · 2024-07-08

**Soundness:** 2
**Presentation:** 2
**Contribution:** 2
**Rating:** 3
**Confidence:** 3

**Summary:**

This work studies human motion prediction in 3D scenes. The proposed DiMoP3D leverages context-aware intermodal interpreter, behaviorally-consistent stochastic planner, and self-prompted motion generator to solve the task. The authors conduct experiments on GIMO and CIRCLE to demonstrate a superior performance.

**Strengths:**

- The paper is well-organized and easy to follow.
- In extensive experiments,  DiMoP3D demonstrates a superior performance over the baselines with analysis.

**Weaknesses:**

- I question the novelty of this work. Novelties in Context Awareness and Autonomous Intention Estimation are overclaimed. They have been studied in previous works but authors fail to provide a discussion:
[29] uses path planning to perform context-aware navigation and GoalNet to estimate the end-point state.
[98] uses 3D scene point to encode the scene context and gaze to estimate the intention.
Still more work to supplement...
I believe the novelty is very overclaimed.


- The system estimates the motion of a fixed length $\Delta L$. Is it possible to not reach an end state? Taking Fig.2. as a case, how do you guarantee the human is sitting on the chair but not still walking to it after $\Delta L$? It may lead to unnature speed and motions.


- How do you evaluate the diversity and natureness of motions?

**Questions:**

See weakness part.

**Limitations:**

As discussed in Appendix G, the system is not as efficient as baselines.

---

> ### Author Rebuttal · Authors · 2024-08-07
>
> **Q1**: The novelty of this work.
>
> **A1**: The task of predicting stochastic human motions with real-world scene awareness is crucial for embodied applications like robotics and autonomous vehicles, enhancing their navigational systems to effectively avoid collisions by considering dynamic real-world conditions. This aspect is usually overlooked by current tasks, and we believe we are the pioneers in introducing scene perception in diversity prediction to achieve better performance in real-world applications. Besides, a novel framework DiMoP3D is proposed to tackle this challenge. Please refer to **A4 to reviewer nnKQ** for details due to the limit of space.
>
> Moreover, compared with BiFU [98]: **(1)** DiMoP3D’s scene interpreter explicitly identifies potential targets as conditional factors for motion generation, in contrast to BiFU’s basic use of PointNet which lacks crossmodal analysis of past motions and intention analysis; **(2)** Human gaze is frequently unavailable in real-world settings, yet it forms the sole basis for inferring human intentions in BiFU. This heavy reliance  significently limits its practicality; **(3)** Different from human gaze which generally indicates deterministic intentions, DiMoP3D analyzes the distribution of human intentions through integrated scene-motion analysis, making it better suited for diverse prediction tasks.
>
> Overall, we will provide a more thorough discussion of our novelty and the differences between the components of DiMoP3D and existing methods in the next version.
>
> **Q2**: The system estimates the motion of a fixed length.
>
> **A2**: For scenarios where the subject reaches the goal within 5 seconds, DiMoP3D sets the subject to remain relatively static upon arrival. If targets are too far to be reached within 5 seconds, our scene interpreter would prevent their selection. In cases where the target is beyond reach within this timeframe, DiMoP3D predicts the initial 5 seconds of motion towards the goal, designating the 'floor' midway as the intended target and 'walk through' as the action of end-pose. Although this issue rarely occurs in the relatively small indoor scenes, it worths further discussion.
>
> The 'floor' is categorized as a special class of object and is segmented by the point segmentator; points from the 'floor' also contribute to the interest calculation. In Equation 4, we calculate the average interest for each object to facilitate sampling of target objects. However, this approach is not suitable for the 'floor' object, as the 'floor' object spread a large area, and the interest levels across different areas of the floor can vary significantly. To address this, we specifically discuss the selection of the 'floor' as a target. We start by calculating the probability of choosing the 'floor' with $\dfrac{\sum_{p \in \text{floor}} \exp(M[p])}{\sum_{p \in S} \exp(M[p])}$. If the 'floor' is selected, a precise target point is identified using GumbelSoftmax within the floor's point cloud. In our experiments in small indoor scenes, the probability of selecting the 'floor' as a target is below 1%; therefore, we have not detailed this in the current version but will include a comprehensive discussion in the next update.
>
> **Q3**: How do you evaluate the diversity and natureness of motions?
>
> **A3**: **For diversity**, we adopt the APD metric (Average Pairwise Distance) following [1,2,3]. It reportes the L2 distances between poses and trajectories of the generated sequences. **For natureness**, we employ the ACPD (Average Cumulated Penetration Depth) to assess scene-motion consistency and physical naturalness, along with the FID (Fréchet Inception Distance) following [4,5,6], which quantifies the discrepancy between the latent distributions of generated motions and ground truth in synthesis. Given the complexity of accurately gauging naturalness, visualization serves as a valuable tool for a more intuitive evaluation.
>
> **Q4**: System efficiency.
>
> **A4**: We argue that the baseline methods are not scene-aware as they do not process 3D scene features, which explains their faster performance. However, DiMoP3D significantly outperforms these baselines, and the trade-off in speed is justifiable given its enhanced capabilities. Additionally, integrating efficient diffusion modules [7,8] and advanced vision modules [9] holds promise for further accelerating DiMoP3D's performance.
>
> [1] Barquero, et al. "Belfusion: Latent diffusion for behavior-driven human motion prediction." ICCV2023.
>
> [2] Mao, et al. "Generating smooth pose sequences for diverse human motion prediction." ICCV2021.
>
> [3] Yuan, et all. "Dlow: Diversifying latent flows for diverse human motion prediction." ECCV2020.
>
> [4] Wang, et al. "Move as You Say Interact as You Can: Language-guided Human Motion Generation with Scene Affordance." CVPR2024.
>
> [5] Tevet, et al. "Human motion diffusion model." ICLR2023.
>
> [6] Wang, et al. "Towards diverse and natural scene-aware 3d human motion synthesis." CVPR2022.
>
> [7] Wang, et al. "Patch diffusion: Faster and more data-efficient training of diffusion models." NIPS2024.
>
> [8] Baranchuk, et al. "Label-efficient semantic segmentation with diffusion models." arXiv preprint arXiv:2112.03126.
>
> [9] Zhu, et al. "Vision mamba: Efficient visual representation learning with bidirectional state space model." arXiv preprint arXiv:2401.09417.

---

> > ### Comment · Reviewer_wnWw · 2024-08-14
> >
> > I thank the authors' responses in their rebuttal. They have addressed my concerns about evaluation and efficiency. I have also read discussions with other reviewers and I share the same concerns about the novelty. I agree a mixture of existing techniques and the mixed task (historical conditions, scene awareness, intention estimation) can be considered as novelty, but somehow incremental. Besides, it is important to review related works with similar motivations faithfully and attribute their contributions and limitations. However, I am not very convinced by their texts. Plus the missing non-trivial details in tackling corner cases, this work is not very convincing to me and not ready for publication. Though I appreciate the authors' efforts in their rebuttal, I still suggest a major revision and can not recommend acceptance at this time.

---

> ### Author Response · Authors · 2024-08-11
> **Comment to Reviewer wnWw from authors**
>
> Dear Reviewer wnWw:
>
> Thank you for the time and effort you've invested in reviewing our manuscript. In our previous response, we addressed your concerns directly and comprehensively. We greatly appreciate your further feedback on our responses and look forward to discussing them with you!

---

> > ### Author Response · Authors · 2024-08-13
> > **Dear Reviewer wnWw**
> >
> > Thank you for your insight to our work! We kindly remind you that the discussion deadline is within 24 hours. We believe that our rebuttal have addressed your concerns, and we are also prepared to answer any further questions you may have.
> >
> > With this in mind, would you consider revising your rating, or if there are further questions or concerns, can we engage in more discussion!

---

> ### Author Response · Authors · 2024-08-14
> **Response to Reviewer wnWw by Authors**
>
> Thank you for your thoughtful feedback on our rebuttal.  We aim to clarify why our work represents a significant contribution to the field.
>
> **Novelty and Impact:**
>
> Our work introduces a new perspective on human motion prediction that is both task-driven and methodologically innovative. While it is true that the proposed framework integrates existing techniques, the way in which these are combined and the novel problems addressed represent a substantial advancement:
>
> - **Task-Oriented Approach:** Our work focuses on a specific challenge—predicting human motions in 3D scenes with real-world awareness, a problem that has not been adequately addressed in previous literature. We argue that *this novel task is a critical step towards enabling intelligent agents to navigate complex environments more safely and effectively*. By incorporating real-world scene information into the motion prediction process, our solution goes beyond merely predicting skeletal poses.
>
> - **Pioneering Approach: Real-World Scene Awareness.** Our work, DiMoP3D, *pioneers the integration of 3D real-world scene awareness*into the inherently stochastic task of human motion prediction. This is a fundamental shift from traditional skeleton-based methods, enhancing the predictive capabilities of AI systems in complex environments.
>
> - **Performance Gains: Enhanced Diversity and Naturalness.** Our framework achieves higher diversity and naturalness in motion predictions, as evidenced by our quantitative evaluations (APD, ACPD, and FID scores) and qualitative visualizations. These improvements are crucial for practical applications where the ability to predict a wide range of plausible human motions is paramount.
>
> **Response to Incremental Concerns:**
>
> We acknowledge that the combination of techniques may appear incremental at first glance, but we contend that the integration itself is non-trivial and leads to a significant leap forward in terms of performance and applicability (as commentator nnKQ notes). Moreover, the specific challenges addressed, **corner cases** (e.g., how to deal with a person arriving early in a prediction window, or not being able to arrive), and the **new challenges of this new task** (e.g., how to simultaneously balance the deterministic nature of the 3D scene given, as well as the stochastic nature of the human body's movements), require careful consideration and innovative solutions.
>
>
> All these considerations suggest that the proposed DiMoP3D possesses enough novelty to go beyond simply incremental progress, especially for this newly proposed task of motion prediction in a 3D real-world point cloud.
>
>
> /* **Remark:**  *We respectfully request that Reviewer wnWw consider the broader implications of our work within the context of human motion prediction in realistic environments. DiMoP3D represents a significant stride in advancing the field towards more human-like AI systems capable of understanding and anticipating behavior in complex, real-world scenarios. We believe this contribution represents a substantial leap forward for the community and the practical application of embodied AI.* */
>
> We are grateful for the opportunity to refine our work based on the feedback provided. Addressing these concerns will not only strengthen our manuscript but also emphasize the importance of DiMoP3D in advancing the field of computer vision and AI towards better human-scene understanding and interaction capabilities.

---

### Official Review · Reviewer_p6AA · 2024-07-12

**Soundness:** 3
**Presentation:** 3
**Contribution:** 3
**Rating:** 6
**Confidence:** 3

**Summary:**

The paper introduces a novel task that incorporates real-world 3D scene information into the existing Human Motion Prediction task. The authors propose a model (DiMoP3D) that, starting from the observed motion sequence and the 3D scene, stochastically predicts the future poses and the interactions with the context; in particular, DiMoP3D infers the individual intention and the human-object interactions. Upon benchmarking on two datasets, GIMO and CIRCLE, the model surpasses all the baselines on most of the considered metrics.

**Strengths:**

The proposed task lies in the intersection of existing ones while capturing a problem setting that hasn’t been explored yet. Similar tasks either considered only the contact points within the scene but with no stochasticity in the prediction ([1,2,3]) or focused on stochastically generating human motion conditioning on the observed pose sequence and the 3D scene but fixing the goal (e.g., [4]). As such, the discussion is technically sound and original. The proposed model is convincingly designed as the role of its submodules is properly motivated and analyzed both in the method’s section and in the ablation studies. The results on CIRCLE and GIMO confirm the effectiveness of the proposal.

[1] Mao, Wei, Richard I. Hartley, and Mathieu Salzmann. "Contact-aware human motion forecasting." Advances in Neural Information Processing Systems 35 (2022): 7356-7367.

[2] Luca Scofano, Alessio Sampieri, Elisabeth Schiele, Edoardo De Matteis, Laura Leal-Taixé, Fabio Galasso. “Staged Contact-Aware Global Human Motion Forecasting.” BMVC 2023: 589-594

[3]  Xing, Chaoyue, Wei Mao, and Miaomiao Liu. "Scene-aware Human Motion Forecasting via Mutual Distance Prediction." arXiv preprint arXiv:2310.00615 (2023).

[4] Hassan, Mohamed, et al. "Stochastic scene-aware motion prediction." Proceedings of the IEEE/CVF International Conference on Computer Vision. 2021.

**Weaknesses:**

W1. While it is clear how their proposed task diverges from human motion synthesis in 3D scenes, human-object and scene interaction prediction (e.g., [5]), and social navigation, it is not clear enough how it differs from scene-aware 3D human motion forecasting or synthesis.

[5] Xu, Sirui, et al. "Interdiff: Generating 3d human-object interactions with physics-informed diffusion." Proceedings of the IEEE/CVF International Conference on Computer Vision. 2023.

**Questions:**

I acknowledge that Sec. B.1 of the supplementary extends the literature review in Sec. 2 of the main manuscript, though the discussion could benefit from a tighter and more direct comparison with similar tasks in [1,2,3,4].

Typos and minor issues with the writing:

L99: forgot the full stop at the end of the sentence.

L119: “intermodal insights. utilizing…”

L181: “To tackle ress these”

L211: “When \alpha_T is approaches 0…”

L261: “...is introduces…”

L304: “...by the subject en route…”

**Limitations:**

Yes, they included the discussion about limitations and societal impact in Section G of the supplementary material.

---

> ### Author Rebuttal · Authors · 2024-08-07
>
> **Q1**: How it differs from scene-aware 3D human motion forecasting or synthesis, and comparisons with similar tasks.
>
> **A1**: **For scene-aware motion forecasting**: Traditional scene-aware 3D human motion forecasting [5,6,7] typically predicts a single sequence based on observation. In contrast, our task models a distribution of potential movements, enabling diverse future scenarios to emerge from a single historical motion. This capability is essential in autonomous vehicles and robotics to effectively anticipate and avoid potential collisions.
> **For scene-aware motion synthesis**: Synthesis methods [8,9,10] generate motions from arbitrary starting positions without considering historical context, while our task requires each generated sequence to be consistent with observed dynamics. Our method contributes to the robotics and navigation, while synthesis methods are often geared towards multimedia and gaming with less constraints.
>
> The mentioned [1,2,3] focus on deterministic prediction, while [4] deals with motion synthesis. We will enhance the discussion of these similar tasks in the next version of our paper.
>
> **Q2**: Typos.
>
> **A2**: We will correct these typos in the next version.
>
> [1] Mao, et al. "Contact-aware human motion forecasting." NIPS2022.
>
> [2] Luca, et al. “Staged Contact-Aware Global Human Motion Forecasting.” BMVC 2023.
>
> [3] Xing, et al. "Scene-aware Human Motion Forecasting via Mutual Distance Prediction." arXiv:2310.00615.
>
> [4] Hassan, Mohamed, et al. "Stochastic scene-aware motion prediction." Proceedings of the IEEE/CVF International Conference on Computer Vision. 2021.
>
> [5] Zheng, et al. "Gimo: Gaze-informed human motion prediction in context." ECCV2022.
>
> [6] Araújo, et al. "Circle: Capture in rich contextual environments." CVPR2023.
>
> [7] Mao, et al. "Contact-aware human motion forecasting." NIPS2022.
>
> [8] Wang, et al. "Humanise: Language-conditioned human motion generation in 3d scenes." NIPS2022.
>
> [9] Hassan, et al. "Stochastic scene-aware motion prediction." ICCV2021.
>
> [10] Huang, et al. "Diffusion-based generation, optimization, and planning in 3d scenes." CVPR2023.

---

> > ### Comment · Reviewer_p6AA · 2024-08-12
> >
> > I thank the authors and the other Reviewers for their insights.
> >
> > I acknowledge that the authors' rebuttal answers my concerns and I appreciate the authors' commitment to further improve the clarity of the discussion of the submitted manuscript.
> >
> > I understand the points that Reviewer nnKQ made: adding more comparisons with methods from similar tasks could clarify more the difference between the proposed task and model and existing works, and why the latter cannot be naively repurposed. However, I disagree with the comment regarding the limited novelty; to the best of my knowledge, the proposed task is indeed part of the paper's novelty and, even if it shares similarities with other tasks, it differs from them likewise the task of human motion forecasting differs from stochastic human motion forecasting or human motion forecasting with contact points. In my opinion, leveraging existing methodologies to tackle a novel problem shouldn't necessarily be regarded as a weakness, since the selection and the adaptation of such methods, if wisely devised, are already a contribution and a starting point to face the new task.

---

> > > ### Author Response · Authors · 2024-08-12
> > > **Reply to Reviewer p6AA from Authors**
> > >
> > > Thank you for your insightful feedback and for recognizing the core ideas and motivations of our work. We are honored to address your concerns and appreciate your alignment with our approach. This article introduces a novel and crucial task in the fields of robotic and autonomous navigation. DiMoP3D is proposed as a solution to this task accordingly, addressing challenges that existing methods cannot overcome.
> > >
> > > If there are any further issues, we are more than willing to discuss them!

---

### Official Review · Reviewer_nnKQ · 2024-07-15

**Soundness:** 2
**Presentation:** 2
**Contribution:** 3
**Rating:** 4
**Confidence:** 5

**Summary:**

This paper introduces a task that is scene-aware diverse human motion prediction. To be specific, given a 3D scene and history motion, this task aims to predict diverse future human motions that are consistent with the scene and history motion. This paper also proposed a model, DiMoP3D, to tackle this task. This model includes a Context-Aware Intermodal Interpreter to encode the scene and find the goal object, a  Behaviorally-Consistent Stochastic Planner to generate the end pose and an obstacle-free trajectory, and a diffusion model to generate the motions on the path.

**Strengths:**

1. This paper tackles an interesting task.
2. The numerical results are good.

**Weaknesses:**

1. As my understanding, if the author follows the original data split of CIRCLE and GIMO, there are no multiple possible future motions for history motion. How to make sure the model can generate diverse future motions trained on such datasets?

2. Please compare the proposed task with [9, 29]. These two works have already tackled the task of scene-aware stochastic prediction. Why do the authors claim they proposed a new task?

3. The author claims that the predictions must adhere to deterministic constraints, including physical consistency. The proposed method only considers the obstacle-free trajectory, but there is no constraint for the generated poses, e.g., there could be penetration with the ground for the walking sequence.

4. What is the insight and novelty of DiMoP3D to make it different from the previous paper? Most techniques are similar to previous work. The scene encoder is from [71], the prediction of diverse goals is similar to [9], the HOI estimation is similar to [1, 2], the trajectory planning is similar to [77, 82, 29], the motion generator is similar to [87].

5. Some parts are hard to read, e.g., the introduction.

6. For the 3D human motion prediction/generation tasks, I prefer to see a video rather than only some figures for the visualization part.

[1] Zhao, Kaifeng, et al. "Compositional human-scene interaction synthesis with semantic control." European Conference on Computer Vision. Cham: Springer Nature Switzerland, 2022.

[2] Hassan, Mohamed, et al. "Populating 3D scenes by learning human-scene interaction." Proceedings of the IEEE/CVF Conference on Computer Vision and Pattern Recognition. 2021.

**Questions:**

The question is the same as Weaknesses. Overall, the paper seems to propose a valuable task and has good results. Addressing the limitations mentioned above would significantly strengthen the work and I am positive to change the score.

**Limitations:**

The authors addressed the limitations.

---

> ### Author Rebuttal · Authors · 2024-08-07
>
> **Q1**: How to generate diverse motions training on such datasets?
>
> **A1**: Similar to [1,2] utilizing single\_history-to-single\_future data, DiMoP3D is designed to model the posterior distribution of potential motions $P(\hat{X}_{L:L+\Delta L} | X\_{1:L}, S)$, rather than a deterministic mapping function $F(\textbf{X}\_{1:L},S)=\textbf{X}\_{L+1:L+\Delta L}$ during training. This approach enables DiMoP3D to predict multiple future sequences by repeatedly sampling from this distribution, facilitating a single-to-multiple mapping.
>
> **Q2**: Other works in scene-aware stochastic prediction.
>
> **A2**: We argue that [9, 29] focus on simplistic, non-realistic scene representations, while our task utilizes real-captured, unstructured, and complex 3D scene point clouds. The methods in [9, 29] cannot adequately address the challenges presented by our configuration, making a direct comparison infeasible.
> **Compare with [9]**: This work processes the scene using a single 2D image and represents observed sequences with 2D skeletons, which significantly limits interaction with real-world 3D scenes. **Compare to task in [29]**: This is a synthesis method rather than prediction. It employs a predefined target object and does not infer intended targets accordingly. Furthermore, while it considers scene features, it lacks the capability for comprehensive perception and understanding necessary for real-world scene-aware motion predictions.
>
> **Q3**: Physical consistency.
>
> **A3**: Our emphasis on physical consistency prioritizes the crossmodal rationality and realism of motions in real-world scenes. While foot penetration is a recognized challenge, it has traditionally been addressed using various techniques [3,4]. After integrating PhysDiff [4] into DiMoP3D, we observed a significant reduction in Average Contact Penetration Depth (ACPD) from 0.98 to 0.56 and a decrease in foot penetration rate from 13.8% to 1.2% in the GIMO dataset. However, this integration relies on external libraries like IsaacGym, which substantially increase computational overhead. Although these methods are complementary to our approach, focusing on effectively avoiding foot penetration remains a promising direction for future research.
>
> **Q4**: What is the insight and novelty of DiMoP3D?
>
> **A4**: The proposed task of real-world scene-aware stochastic motion prediction is novel yet important to the field of autonomious vechiles and robotics. Stochastic prediction is essential for navigation applications to prevent collisions, yet existing methods fail to account for real-world scenes, yielding suboptimal results. To the best of our knowledge, we are the first to propose this task and develop DiMoP3D to address it. The components in DiMoP3D significantly diverges from existing approaches in several key ways:
>
> **Scene interpreter**: Our interpreter innovatively proposes to explicitly predict potential target objects through crossmodal analysis of past motion and the scene. The scene encoder [71] processes the point cloud to map it into the motion feature space, with target estimation remaining the focus of our interpreter. Besides, although [9] predicts diverse goals, it only identifies pixel points from simple 2D images, failing to recognize interactive objects with low accuracy, and thus does not effectively capture motion intentions in real 3D scenes.
>
> **Trajectory planning**: We plan stochastic obstacle-free trajectories towards targets in real-world scenes, addressing shortcomings of previous methods: **(1)** [29] uses traditional A* that generates deterministic trajectories, contrasting with our diverse prediction strategy. **(2)** [77] treats the scene as a binary map, where any position or point with a height greater than zero is considered an obstacle and thus blocked. In contrast, we recognize that some obstacles in real-world scenes are partially traversable and develope a numerically-continuous scene map (refer to Table 7). **(3)** [82] applies a module-based controller to navigate, which struggles with direct adaptation to real-world 3D point clouds and is challenging to fine-tune due to data limitations.
>
> **Human estimator**: While our estimator is inspired by previous research, its application is distinct. In our model, it serves as a deterministic semantic constraint, guiding the motion generator to comply with the scene context. This integration is part of our approach to tackling the challenge of scene-aware diverse motion prediction.
>
> **Motion generator**: The main insight of our motion generator is to harmonize the stochasticity of human movements with the deterministic constraints of the real-world scenes. Unlike [87], which only considers deterministic constraints between the contact joint (e.g., hand) and a single object, our approach accounts for whole-body constraints within the entire real-world 3D scene, guided by a self-prompted stochastic factor.
>
> **Q5**: Some parts are hard to read.
>
> **A5**: The introduction comprises five paragraphs: (1) the importance of stochastic motion prediction; (2) the limited consideration of real-world scenes in existing stochastic prediction methods; (3) the key challenges associated with scene-aware stochastic motion prediction; (4) the structure of our DiMoP3D, designed to address these challenges; (5) our contributions. We will refine the structure of the article to enhance readability and clarity in the next version. Welcome for any additional feedback!
>
> **Q6**: Video samples.
>
> **A6**: Kindly refer to the anonymous page in Appendix. A (line-588).
>
> [1] Barquero, et al. "Belfusion: Latent diffusion for behavior-driven human motion prediction." ICCV2023.
>
> [2] Yuan, et al. "Dlow: Diversifying latent flows for diverse human motion prediction." ECCV2020.
>
> [3] Liu, et al. "Learning basketball dribbling skills using trajectory optimization and deep reinforcement learning." TOG2018.
>
> [4] Ye, et al. "Physdiff: Physics-guided human motion diffusion model." ICCV2023.

---

> > ### Comment · Reviewer_nnKQ · 2024-08-11
> >
> > Thanks for the author's clarification!
> >
> > Q1: I understand the authors are trying to model the posterior distribution of potential motions. However, for the stochastic human motion prediction task, the posterior distribution is $P(X_{L:L+\Delta{L}}|X_{1:L})$ and there are many motions with similar history but different futures in the dataset, allowing them to learn a good distribution. But for the scene-aware stochastic human motion prediction task, the posterior distribution is $P(X_{L:L+\Delta{L}}|X_{1:L}, S)$ and there are very few motions with similar history but different futures in the same scene.
> >
> > Q2: For [29], what is the key difference? And why does it lack such a capability?
> >
> > Q3: Thank you for your experiments. However, the scene in this task is considerably more complex than in PhysDiff. The generated motion is not constrained solely by foot placement; other factors must also be considered. For example, there might be cases where your generated motion results in penetration with objects like tables or chairs when sitting down. Walking and penetrating the ground is just one scenario among many.
> >
> > Q4: In my opinion, scene-aware stochastic motion prediction is distinctively conditioned on historical motion, setting it apart from other scene-aware synthesis methods. Therefore, it is equally important to compare it with these scene-aware synthesis methods rather than solely focusing on stochastic motion prediction. While the proposed framework is well-structured, it builds upon and integrates existing methodologies, leading to a limited novelty. The primary contribution seems to lie in the Scene Interpreter, which predicts potential target objects from 3D point clouds, as opposed to the previous work [9] that predicted targets from images.
> >
> >
> > Q5: Thanks for your clarification.
> >
> > Q6: There are only comparisons with BelFusion. A comparison with scene-aware methods is also important.
> >
> > Q7: I acknowledge that your results will likely surpass those of BelFusion since you incorporate scene information, whereas BelFusion does not. However, I'm more curious about the comparison between your method and other scene-aware approaches. For instance, you could easily adapt language-conditioned scene-aware synthesis methods by replacing the condition from language to history motion. What insights do you have that demonstrate your method's superiority over previous scene-aware methods? In what cases do previous methods fall short, and how does your approach address these limitations?

---

> ### Author Response · Authors · 2024-08-11
> **Reply to Reviewer nnKQ from Authors**
>
> **A1**: We appreciate the reviewer's point highlighting the challenging of our task as compared to traditional stochastic prediction. DiMoP3D overcomes this by decomposing the problem into target prediction, interactive pose estimation, path planning, and self-prompted motion generation that handle different aspects of the scene-aware diverse motion prediction. Thus, $ P(\hat{X}_{L:L+\Delta L} | X\_{1:L}, S) $ could be represented as the product of terms: $P(O_g | X\_{1:L}, S)$, $P(\hat{X}\_{L+\Delta L} | X\_{1:L}, S, O_g)$,  and $P(\hat{X}\_{L:L+\Delta L} | X\_{1:L}, \hat{X}\_{L+\Delta L}, \hat{\tau}^{plan}, S)$. **(1)** Our interpreter learns the distribution rules of potential targets within scenes $P(O_g | X\_{1:L}, S)$. For instance, a subject traversing an aisle is modeled to likely interact with objects along that path, while a subject facing a chair is likely to sit. This module capitalizes on patterns in the relationship between past motion and scene context to predict potential targets, effectively modeling the target distribution despite the limited frequency of specific motion histories. **(2)** Pose estimator models how interactive poses vary with different objects within the scene $P(\hat{X}\_{L+\Delta L} | X\_{1:L}, S, O_g)$. while similar motions in different scenes might lead to different interactions, the core interaction with objects (like sitting or picking) remains consistent across contexts. **(3)** Motion generator models coherent human movement based on learned stochastic factors $P(\hat{X}\_{L:L+\Delta L} | X\_{1:L}, \hat{X}\_{L+\Delta L}, \hat{\tau}^{plan}, S)$. The distribution of walking movements and the coherent switching of different poses is rich in our scene-aware data, making it easy to learn.
>
> **A2**: **(1)** They ultilize pre-defined targets, meaning that the target object and location is fixed and known before the prediction begins, while we have to predict the distribution of potential targets dynamically by analyzing past motion and the scene context. **(2)** The scene-awareness of [29] is only based on the 3D voxel grid (8x8x8) of the pre-defined target and planning deterministic paths. In the contrary, DiMoP3D engages with the entire 3D scene, incorporating detailed, real-world environmental data. **(3)** They operate within a single, manually synthetic scene, where all objects' shapes, positions and features are known to the system, eliminating the need for the module to learn. In real-world applications, the ground truth of the scene and object is unaccessible, limiting its use. However, our DiMoP3D is designed to recoginize and analyze potential targets in a variety of real-world scenes.
>
> **A3**: Incorporating the Signed Distance Field (SDF) of the human mesh and scene point cloud into the reinforcement learning module’s reward function may help avoiding issues like object penetration. While this approach is promising, the complexity of integrating it with real-world 3D scene data is substantial. Given that it is not the core focus of this work, we leave it in future research, which is a promising direction.
>
> **A4**: For comparison to scene-aware synthesis, kindly refer to Appendix. B. We emphasise that DiMoP3D significantly adapts existing methods to the novel task of scene-aware diverse motion prediction. This is not merely an incremental improvement; it addresses a previously unexplored task. We believe our method represents a significant leap in the field, pioneering a new and promising track for motion prediction in real-world scenarios.
>
> **A5**: Welcome for any additional feedback!
>
> **A6**: We emphasize that our task diverges from scene-aware synthesis and deterministic prediction. Nonetheless, for the sake of completeness, we compare our approach with scene-aware synthesis in Appendix B. Additionally, we provide a supplemental visualization comparison with scene-aware BiFU in Section 5 on our anonymous page (line 588 in Appendix A).
>
> **A7** Even with the last observed frame and embedding of past motion incorporated, synthesis methods still struggle to predict future motions coherently (especially switching between observation and prediction). In time-series predictive tasks, history should not be treated as a conditional factor but as a foundational element that future sequences must adhere to. Treating historical information as a condition ignores the temporal correlation between history and the future, while using it as a fundamental element means that the predicted results must strictly maintain consistency with historical information. This is the underlying reason why using historical information as a condition for LLM (or the synthesis methods) cannot achieve high fidelity prediction. Appendix B provides specific qualitative experiments, in which the discontinuity between historical and predicted actions in the comparative method (AffordMotion, CVPR'24) provides evidence for our statement.
>
> **We are looking forward for further discussions!**

---

> ### Comment · Reviewer_nnKQ · 2024-08-12
>
> My main concerns, specifically regarding Q4 and Q7, have not yet been addressed.
>
> For Q4, the novelty and insight of the proposed method remain underwhelming. 'While the proposed framework is well-structured, it builds upon and integrates existing methodologies, leading to a limited novelty. The primary contribution seems to lie in the Scene Interpreter, which predicts potential target objects from 3D point clouds, as opposed to the previous work [9] that predicted targets from images.'
>
> For Q7, although the results are good, it lack a compelling argument. Your reasoning is that "In time-series predictive tasks, history should not be treated as a conditional factor but as a foundational element that future sequences must adhere to. Treating historical information as a condition overlooks the temporal correlation between history and the future, whereas using it as a fundamental element ensures that the predicted results strictly maintain consistency with historical information." However, this appears to be just a different design choice for the diffusion process. Diffusion-based scene-aware synthesis methods could also adopt this approach, such as [1, 2], to predict future frames by infilling ground truth past motion into the denoised motion at each diffusion step. On the other hand, Interdiff [3] observes that encoding the historical motion as a condition leads to better performance. I do not see any limitation here. Please review my questions carefully.
>
> I believe the key difference between scene-aware human motion synthesis methods and stochastic scene-aware human motion prediction lies in the condition. Please provide a robust argument regarding this distinction for your proposed methods.
>
> [1] Ye Yuan, Jiaming Song, Umar Iqbal, Arash Vahdat, and Jan Kautz. PhysDiff: Physics-guided human motion diffusion model. In ICCV, 2023.
>
> [2] Mingyuan Zhang, Zhongang Cai, Liang Pan, Fangzhou Hong, Xinying Guo, Lei Yang, and Ziwei Liu. MotionDiffuse: Text-driven human motion generation with diffusion model. arXiv preprint arXiv:2208.15001, 2022. 3, 4
>
> [3] Xu, Sirui, et al. "Interdiff: Generating 3d human-object interactions with physics-informed diffusion." Proceedings of the IEEE/CVF International Conference on Computer Vision. 2023.

---

> ### Author Response · Authors · 2024-08-12
> **Reply to Reviewer nnKQ from Authors**
>
> **A7**: Thank you for your insights. While treating historical motion as a condition has proven effective in certain synthesis methods, it poses significant limitations in the context of diverse prediction:
>
> **(1) Lack of crossmodal analysis between past motion and scene**. Even when incorporating and infilling the observed sequence into diffusion-based synthesis methods, they often lack an explicit crossmodal analysis between past motion and the scene. This analysis is crucial for inferring potential human intentions and goes beyond traditional scene-motion attention in synthesis methods.
>
> **(2) Difficulty in reconstructing observation**. To ensure consistency between the prediction and observation, the predictor in [3] is forced to predict observation as well (train_diffusion_skeleton.py, line-190 in official github repo of InterDiff). However, accurately reconstructing observation is also challenging. During inference, the predicted observation part (0.33sec out of 1.07sec in total in InterDiff) is in fact apart from ground truth. In our long-term prediction task with a 3-second observation phase, this method would struggle to reconstruct long-term observations accurately, which negatively impacts performance.
>
> **(3) Motion incoherence**. For methods that not infilling observation during inference, there may be issues in motion incoherence between observed and predicted sequences, as noticed by [4,5]. As a remedial measure, [4] proposes adversarial training to force the module to reconstruct the sequence (including observation) from the past motion embedding; [5] employs a post-refinement method to align the sequences better. These methods are more complex and introducing additional computational overhead.
>
> **(4) Posterior collapse**. [5] finds that encoding past motion as a condition can cause posterior collapse during joint training, as strong decoders may ignore the learned latent variables. DiMoP3D overcomes this by decomposing the task into sub-tasks, making each sub-module easy to learn, ensuring both high fidelity and diversity even with limited data.
>
> **(5) Scene-awareness**. There are limited scene-aware synthesis methods, most of them (including [1,2]) lack scene awareness. The SoTA diffusion-based scene-aware synthesis method AffordMotion(CVPR24) [6], discussed in Appendix B, has limitations: **(a) past motion embedding** causes significant motion incoherence (see Appendix B.3), and **(b) observation infilling** may lead to significant posterior collapse.
>
> We will incorporates more detailed dicussion and experimental comparison with synthesis methods in the next version!
>
> **A4**: The primary novelty of our work lies in introducing the task of diverse motion prediction in real-world 3D scenarios, a challenge not addressed by current methods, as confirmed by Reviewer p6AA. Our approach deconstructs the challenge and adapts advanced methods from the community to fit this unique context. Significant enhancements have been made to existing technologies, including Scene Interpreter (crossmodal analysis), Path Planner (efficient stochastic navigation), and Motion Generator (self-prompted on the end-pose and trajectory). We kindly request that you reevaluate our contributions and novelty from this perspective, considering our comprehensive approach to addressing the complexities of diverse scene-aware motion prediction.
>
> [1] Ye Yuan, Jiaming Song, Umar Iqbal, Arash Vahdat, and Jan Kautz. PhysDiff: Physics-guided human motion diffusion model. In ICCV, 2023.
>
> [2] Mingyuan Zhang, Zhongang Cai, Liang Pan, Fangzhou Hong, Xinying Guo, Lei Yang, and Ziwei Liu. MotionDiffuse: Text-driven human motion generation with diffusion model. arXiv preprint arXiv:2208.15001, 2022. 3, 4
>
> [3] Xu, Sirui, et al. "Interdiff: Generating 3d human-object interactions with physics-informed diffusion." Proceedings of the IEEE/CVF International Conference on Computer Vision. 2023.
>
> [4] Barquero, German, Sergio Escalera, and Cristina Palmero. "Belfusion: Latent diffusion for behavior-driven human motion prediction." Proceedings of the IEEE/CVF International Conference on Computer Vision. 2023.
>
> [5] Wei, Dong, et al. "Human joint kinematics diffusion-refinement for stochastic motion prediction." Proceedings of the AAAI Conference on Artificial Intelligence. Vol. 37. No. 5. 2023.
>
> [6] Wang, Zan, et al. "Move as You Say Interact as You Can: Language-guided Human Motion Generation with Scene Affordance." Proceedings of the IEEE/CVF Conference on Computer Vision and Pattern Recognition. 2024.

---

> > ### Author Response · Authors · 2024-08-13
> > **Dear Reviewer nnKQ**
> >
> > Thank you for your interest in our work, your prompt responses, and your thorough engagement with us! We kindly remind you that the discussion deadline is within 24 hours. We trust that our comprehensive responses regarding synthesis tasks and methods have addressed your concerns. We are also prepared to answer any further questions you may have.
> >
> > With this in mind, would you consider revising your rating, or if there are further questions or concerns, can we engage in more discussion!

---

> > > ### Comment · Reviewer_nnKQ · 2024-08-14
> > >
> > > I appreciate the author's rebuttal. After thoroughly reviewing the paper, supplementary materials, rebuttal, and comments from other reviewers, I have decided to maintain my rating as borderline reject based on the current quality of the work. While I recognize the contribution of the proposed new task, the contribution of the method remains limited, and the author's arguments have not convinced me. I concur with reviewer wnWw on the importance of faithfully reviewing related works with similar motivations, properly attributing their contributions and acknowledging their limitations. I recommend the author address these concerns in the main paper to improve the overall quality of the work.

---

### Decision · Program_Chairs · 2024-09-25

**Decision:**

Accept (poster)

**Comment:**

This paper proposes the novel task of predicting diverse human motion in real-world 3D scenes, therefore abiding by stochastic predictions and 3D-scene-conditioning, to test with proposed metrics in the datasets of GIMO and CIRCLE and a new model for the task termed DiMoP3D.

Upon the rebuttal and the authors-reviewers discussion, the reviews were diverging, and one was borderline: 1x reject, 1x borderline reject, 1x weak accept, and 1x strong accept.

The AC summarizes these main aspects:

(1) The paper is clear and well-motivated, as agreed by most reviewers and confirmed by the AC

(2) The performance of the proposed model is effective, and the selected baselines are sensible, as agreed by all reviewers and confirmed by the AC.

(3) The proposed task is novel. Two reviewers are convinced about this aspect. Two reviewers are not convinced of this. The AC notes that the discussion with related work needs to be improved. The many raised questions clearly show that the authors need to make clearer points on the novelty and how this task differs from the existing ones. Effectively, the AC understands the two contrary reviewers' viewpoints, as this task appears incremental with respect to those tasks and benchmarks that are currently available. On the other hand, the AC stands on the positive side, arguing that proposing to bring together stochasticity and 3D real scenes is new and a great direction to proceed in this research field.

(4) The modules employed in DiMoP3D are mostly derived from best practices with limited novelty.

(5) As proposed, the new task is deemed helpful in progressing the relevant application fields. The proposed datasets and metrics appear to be an appropriate step forward. Progress on this task will likely reflect in progress across all the existing compound subtasks. This paper is the first to have proposed the task of diverse scene-aware motion prediction. The selected datasets are recent and large enough to ignite follow-up work on the task. The metrics are best practices in the field of scene-aware and stochastic human motion prediction, so the performance depicted in the paper is representative of modeling progress.

Further to the notes on novelty and performance, the motivation stems from the vast utility and timely need of the proposed task for application fields of significant potential impact, such as human-robot-scene navigation and interaction. As such, we recommend acceptance of this paper.